# Regret bounds for meta Bayesian optimization with an unknown Gaussian process prior

**Zi Wang***
MIT CSAIL
ziw@csail.mit.edu

**Beomjoon Kim***
MIT CSAIL
beomjoon@mit.edu

**Leslie Pack Kaelbling**
MIT CSAIL
lpk@csail.mit.edu

## Abstract

Bayesian optimization usually assumes that a Bayesian prior is given. However, the strong theoretical guarantees in Bayesian optimization are often regrettably compromised in practice because of unknown parameters in the prior. In this paper, we adopt a variant of empirical Bayes and show that, by estimating the Gaussian process prior from offline data sampled from the same prior and constructing unbiased estimators of the posterior, variants of both GP-UCB and *probability of improvement* achieve a near-zero regret bound, which decreases to a constant proportional to the observational noise as the number of offline data and the number of online evaluations increase. Empirically, we have verified our approach on challenging simulated robotic problems featuring task and motion planning.

## 1 Introduction

Bayesian optimization (BO) is a popular approach to optimizing black-box functions that are expensive to evaluate. Because of expensive evaluations, BO aims to approximately locate the function maximizer without evaluating the function too many times. This requires a good strategy to adaptively choose where to evaluate based on the current observations.

BO adopts a Bayesian perspective and assumes that there is a prior on the function; typically, we use a Gaussian process (GP) prior. Then, the information collection strategy can rely on the prior to focus on good inputs, where the goodness is determined by an acquisition function derived from the GP prior and current observations. In past literature, it has been shown both theoretically and empirically that if the function is indeed drawn from the given prior, there are many acquisition functions that BO can use to locate the function maximizer quickly [51, 5, 53].

However, in reality, the prior we choose to use in BO often does not reflect the distribution from which the function is drawn. Hence, we sometimes have to estimate the hyper-parameters of a chosen form of the prior *on the fly* as we collect more data [50]. One popular choice is to estimate the prior parameters using empirical Bayes with, e.g., the maximum likelihood estimator [44] .

Despite the vast literature that shows many empirical Bayes approaches have well-founded theoretical guarantees such as consistency [40] and admissibility [26], it is difficult to analyze a version of BO that uses empirical Bayes because of the circular dependencies between the estimated parameters and the data acquisition strategies. The requirement to select the prior model and estimate its parameters leads to a BO version of the chicken-and-egg dilemma: the prior model selection depends on the data collected and the data collection strategy depends on having a "correct" prior. Theoretically, there is little evidence that BO with unknown parameters in the prior can work well. Empirically, there is evidence showing it works well in some situations, but not others [33, 23], which is not surprising in light of no free lunch results [56, 22].

---

In this paper, we propose a simple yet effective strategy for learning a prior in a meta-learning setting where training data on functions from the same Gaussian process prior are available. We use a variant of empirical Bayes that gives unbiased estimates for both the parameters in the prior and the posterior given observations of the function we wish to optimize. We analyze the regret bounds in two settings: (1) finite input space, and (2) compact input space in $\mathbb{R}^d$. We clarify additional assumptions on the training data and form of Gaussian processes of both settings in Sec. 4.1 and Sec. 4.2. We prove theorems that show a near-zero regret bound for variants of GP-UCB [2, 51] and *probability of improvement* (PI) [29, 53]. The regret bound decreases to a constant proportional to the observational noise as online evaluations and offline data size increase.

From a more pragmatic perspective on Bayesian optimization for important areas such as robotics, we further explore how our approach works for problems in task and motion planning domains [27], and we explain why the assumptions in our theorems make sense for these problems in Sec. 5. Indeed, assuming a common kernel, such as squared exponential or Matérn, is very limiting for robotic problems that involve discontinuity and non-stationarity. However, with our approach of setting the prior and posterior parameters, BO outperforms all other methods in the task and motion planning benchmark problems.

The contributions of this paper are (1) a stand-alone BO module that takes in only a multi-task training data set as input and then actively selects inputs to efficiently optimize a new function and (2) analysis of the regret of this module. The analysis is constructive, and determines appropriate hyperparameter settings for the GP-UCB acquisition function. Thus, we make a step forward to resolving the problem that, despite being used for hyperparameter tuning, BO algorithms themselves have hyperparameters.

## 2   Background and related work

**BO** optimizes a black-box objective function through sequential queries. We usually assume knowledge of a Gaussian process [44] prior on the function, though other priors such as Bayesian neural networks and their variants [17, 30] are applicable too. Then, given possibly noisy observations and the prior distribution, we can do Bayesian posterior inference and construct acquisition functions [29, 38, 2] to search for the function optimizer.

However, in practice, we do not know the prior and it must be estimated. One of the most popular methods of prior estimation in BO is to optimize mean/kernel hyper-parameters by maximizing data-likelihood of the current observations [44, 19]. Another popular approach is to put a prior on the mean/kernel hyper-parameters and obtain a distribution of such hyper-parameters to adapt the model given observations [20, 50]. These methods require a predetermined form of the mean function and the kernel function. In the existing literature, mean functions are usually set to be 0 or linear and the popular kernel functions include Matérn kernels, Gaussian kernels, linear kernels [44] or additive/product combinations of the above [11, 24].

**Meta BO** aims to improve the optimization of a given objective function by learning from past experiences with other similar functions. Meta BO can be viewed as a special case of transfer learning or multi-task learning. One well-studied instance of meta BO is the machine learning (ML) hyper-parameter tuning problem on a dataset, where, typically, the validation errors are the functions to optimize [14]. The key question is how to transfer the knowledge from previous experiments on other datasets to the selection of ML hyper-parameters for the current dataset.

To determine the similarity between validation error functions on different datasets, meta-features of datasets are often used [6]. With those meta-features of datasets, one can use contextual Bayesian optimization approaches [28] that operate with a probabilistic functional model on both the dataset meta-features and ML hyper-parameters [3]. Feurer et al. [16], on the other hand, used meta-features of datasets to construct a distance metric, and to sort hyper-parameters that are known to work for similar datasets according to their distances to the current dataset. The best k hyper-parameters are then used to initialize a vanilla BO algorithm. If the function meta-features are not given, one can estimate the meta-features, such as the mean and variance of all observations, using Monte Carlo methods [52], maximum likelihood estimates [57] or maximum *a posteriori* estimates [43, 42].

As an alternative to using meta-features of functions, one can construct a kernel between functions. For functions that are represented by GPs, Malkomes et al. [36] studied a "kernel kernel", a kernel for kernels, such that one can use BO with a "kernel kernel" to select which kernel to use to model or

optimize an objective function [35] in a Bayesian way. However, [36] requires an initial set of kernels to select from. Instead, Golovin et al. [18] introduced a setting where the functions come in sequence and the posterior of the former function becomes the prior of the current function. Removing the assumption that functions come sequentially, Feurer et al. [15] proposed a method to learn an additive ensemble of GPs that are known to fit all of those past "training functions".

Theoretically, it has been shown that meta BO methods that use information from similar functions may result in an improvement for the cumulative regret bound [28, 47] or the simple regret bound [42] with the assumptions that the GP priors are given. If the form of the GP kernel is given and the prior mean function is 0 but the kernel hyper-parameters are unknown, it is possible to obtain a regret bound given a range of these hyper-parameters [54]. In this paper, we prove a regret bound for meta BO where the GP prior is unknown; this means, neither the range of GP hyper-parameters nor the form of the kernel or mean function is given.

A more ambitious approach to solving meta BO is to train an end-to-end system, such as a recurrent neural network [21], that takes the history of observations as an input and outputs the next point to evaluate [8]. Though it has been demonstrated that the method in [8] can learn to trade-off exploration and exploitation for a short horizon, it is unclear how many "training instances", in the form of observations of BO performed on similar functions, are necessary to learn the optimization strategies for any given horizon of optimization. In this paper, we show both theoretically and empirically how the number of "training instances" in our method affects the performance of BO.

Our methods are most similar to the BOX algorithm [27], which uses evaluations of previous functions to make point estimates of a mean and covariance matrix on the values over a discrete domain. Our methods for the discrete setting (described in Sec. 4.1) directly improve on BOX by choosing the exploration parameters in GP-UCB more effectively. This general strategy is extended to the continuous-domain setting in Sec. 4.2, in which we extend a method for learning the GP prior [41] and the use the learned prior in GP-UCB and PI.

**Learning how to learn**, or "meta learning", has a long history in machine learning [46]. It was argued that learning how to learn is "learning the prior" [4] with "point sets" [37], a set of iid sets of potentially non-iid points. We follow this simple intuition and present a meta BO approach that learns its GP prior from the data collected on functions that are assumed to have been drawn from the same prior distribution.

**Empirical Bayes** [45, 26] is a standard methodology for estimating unknown parameters of a Bayesian model. Our approach is a variant of empirical Bayes. We can view our computations as the construction of a sequence of estimators for a Bayesian model. The key difference from traditional empirical Bayes methods is that we are able to prove a regret bound for a BO method that uses estimated parameters to construct priors and posteriors. In particular, we use frequentist concentration bounds to analyze Bayesian procedures, which is one way to certify empirical Bayes in statistics [49, 13].

## 3   Problem formulation and notations

Unlike the standard BO setting, we do not assume knowledge of the mean or covariance in the GP prior, but we do assume the availability of a dataset of iid sets of potentially non-iid observations on functions sampled from the same GP prior. Then, given a new, unknown function sampled from that same distribution, we would like to find its maximizer.

More formally, we assume there exists a distribution $GP(\mu, k)$, and both the mean $\mu : \mathfrak{X} \to \mathbb{R}$ and the kernel $k : \mathfrak{X} \times \mathfrak{X} \to \mathbb{R}$ are unknown. Nevertheless, we are given a dataset $\bar{D}_N = \{[(\bar{x}_{ij}, \bar{y}_{ij})]_{j=1}^{M_i}\}_{i=1}^{N}$, where $\bar{y}_{ij}$ is drawn independently from $\mathcal{N}(f_i(\bar{x}_{ij}), \sigma^2)$ and $f_i : \mathfrak{X} \to \mathbb{R}$ is drawn independently from $GP(\mu, k)$. The noise level $\sigma$ is unknown as well. We will specify inputs $\bar{x}_{ij}$ in Sec. 4.1 and Sec. 4.2.

Given a new function $f$ sampled from $GP(\mu, k)$, our goal is to maximize it by sequentially querying the function and constructing $D_T = [(x_t, y_t)]_{t=1}^{T}$, $y_t \sim \mathcal{N}(f(x_t), \sigma^2)$. We study two evaluation criteria: (1) the *best-sample simple regret* $r_T = \max_{x \in \mathfrak{X}} f(x) - \max_{t \in [T]} f(x_t)$ which indicates the value of the best query in hindsight, and (2) the *simple regret*, $R_T = \max_{x \in \mathfrak{X}} f(x) - f(\hat{x}_T^*)$ which measures how good the inferred maximizer $\hat{x}_T^*$ is.

**Notation** We use $\mathcal{N}(u, V)$ to denote a multivariate Gaussian distribution with mean $u$ and variance $V$ and use $\mathcal{W}(V, n)$ to denote a Wishart distribution with $n$ degrees of freedom and scale matrix $V$. We also use $[n]$ to denote $[1, \cdots, n], \forall n \in \mathbb{Z}^+$. We overload function notation for evaluations on vectors $\boldsymbol{x} = [x_i]_{i=1}^n, \boldsymbol{x}' = [x_j]_{j=1}^{n'}$ by denoting the output column vector as $\mu(\boldsymbol{x}) = [\mu(x_i)]_{i=1}^n$, and the output matrix as $k(\boldsymbol{x}, \boldsymbol{x}') = [k(x_i, x_j')]_{i\in[n], j\in[n']}$, and we overload the kernel function $k(\boldsymbol{x}) = k(\boldsymbol{x}, \boldsymbol{x})$.

## 4 Meta BO and its theoretical guarantees

Instead of hand-crafting the mean $\mu$ and kernel $k$, we estimate them using the training dataset $\bar{D}_N$. Our approach is fairly straightforward: in the offline phase, the training dataset $\bar{D}_N$ is collected and we obtain estimates of the mean function $\hat{\mu}$ and kernel $\hat{k}$; in the online phase, we treat $GP(\hat{\mu}, \hat{k})$ as the Bayesian "prior" to do Bayesian optimization. We illustrate the two phases in Fig. 1. In Alg. 1, we depict our algorithm, assuming the dataset $\bar{D}_N$ has been collected. We use ES-TIMATE($\bar{D}_N$) to denote the "prior" estimation and INFER($D_t; \hat{\mu}, \hat{k}$) the "posterior" inference, both of which we will introduce in Sec. 4.1 and Sec. 4.2. For acquisition functions, we consider special cases of *probability of improvement* (PI) [53, 29] and *upper confidence bound* (GP-UCB) [51, 2]:

---

**Algorithm 1** Meta Bayesian optimization

1: **function** META-BO($\bar{D}_N, f$)
2:     $\hat{\mu}(\cdot), \hat{k}(\cdot, \cdot) \leftarrow$ ESTIMATE($\bar{D}_N$)
3:     **return** BO($f, \hat{\mu}, \hat{k}$)
4: **end function**

5: **function** BO ($f, \hat{\mu}, \hat{k}$)
6:     $D_0 \leftarrow \emptyset$
7:     **for** $t = 1, \cdots, T$ **do**
8:         $\hat{\mu}_{t-1}(\cdot), \hat{k}_{t-1}(\cdot) \leftarrow$ INFER($D_{t-1}; \hat{\mu}, \hat{k}$)
9:         $\alpha_{t-1}(\cdot) \leftarrow$ ACQUISITION ($\hat{\mu}_{t-1}, \hat{k}_{t-1}$)
10:        $x_t \leftarrow \arg\max_{x\in\mathfrak{X}} \alpha_{t-1}(x)$
11:        $y_t \leftarrow$ OBSERVE($f(x_t)$)
12:        $D_t \leftarrow D_{t-1} \cup [(x_t, y_t)]$
13:    **end for**
14:    **return** $D_T$
15: **end function**

---

$$\alpha_{t-1}^{\text{PI}}(x) = \frac{\hat{\mu}_{t-1}(x) - \hat{f}^*}{\hat{k}_{t-1}(x)^{\frac{1}{2}}}, \quad \alpha_{t-1}^{\text{GP-UCB}}(x) = \hat{\mu}_{t-1}(x) + \zeta_t \hat{k}_{t-1}(x)^{\frac{1}{2}}.$$

Here, PI assumes additional information[2] in the form of the upper bound on function value $\hat{f}^* \geq \max_{x\in\mathfrak{X}} f(x)$. For GP-UCB, we set its hyperparameter $\zeta_t$ to be

$$\zeta_t = \frac{\left(6(N - 3 + t + 2\sqrt{t\log\frac{6}{\delta}} + 2\log\frac{6}{\delta})/(\delta N(N - t - 1))\right)^{\frac{1}{2}} + (2\log(\frac{3}{\delta}))^{\frac{1}{2}}}{(1 - 2(\frac{1}{N-t}\log\frac{6}{\delta})^{\frac{1}{2}})^{\frac{1}{2}}},$$

where $N$ is the size of the dataset $\bar{D}_N$ and $\delta \in (0, 1)$. With probability $1 - \delta$, the regret bound in Thm. 2 or Thm. 4 holds with these special cases of GP-UCB and PI. Under two different settings of the search space $\mathfrak{X}$, finite $\mathfrak{X}$ and compact $\mathfrak{X} \in \mathbb{R}^d$, we show how our algorithm works in detail and why it works via regret analyses on the best-sample simple regret. Finally in Sec. 4.3 we show how the simple regret can be bounded. **The proofs of the analyses can be found in the appendix.**

### 4.1 $\mathfrak{X}$ is a finite set

We first study the simplest case, where the function domain $\mathfrak{X} = [\bar{x}_j]_{j=1}^M$ is a finite set with cardinality $|\mathfrak{X}| = M \in \mathbb{Z}^+$. For convenience, we treat this set as an ordered vector of items indexed by $j \in [M]$. We collect the training dataset $\bar{D}_N = \{[(\bar{x}_j, \bar{\delta}_{ij}\bar{y}_{ij})]_{j=1}^M\}_{i=1}^N$, where $\bar{y}_{ij}$ are independently drawn from $\mathcal{N}(f_i(\bar{x}_j), \sigma^2)$, $f_i$ are drawn independently from $GP(\mu, k)$ and $\bar{\delta}_{ij} \in \{0, 1\}$. Because the training data can be collected offline by querying the functions $\{f_i\}_{i=1}^N$ in parallel, it is not unreasonable to assume that such a dataset $\bar{D}_N$ is available. If $\bar{\delta}_{ij} = 0$, it means the $(i, j)$-th entry of the dataset $\bar{D}_N$ is missing, perhaps as a result of a failed experiment.

**Estimating GP parameters** If $\bar{\delta}_{ij} < 1$, we have missing entries in the observation matrix $\bar{Y} = [\bar{\delta}_{ij}\bar{y}_{ij}]_{i\in[N],j\in[M]} \in \mathbb{R}^{N\times M}$. Under additional assumptions specified in [7], including that $\text{rank}(Y) = r$ and the total number of valid observations $\sum_{i=1}^{N}\sum_{j=1}^{M}\bar{\delta}_{ij} \geq O(rN^{\frac{6}{5}}\log N)$, we can use matrix completion [7] to fully recover the matrix $\bar{Y}$ with high probability. In the following, we proceed by considering completed observations only.

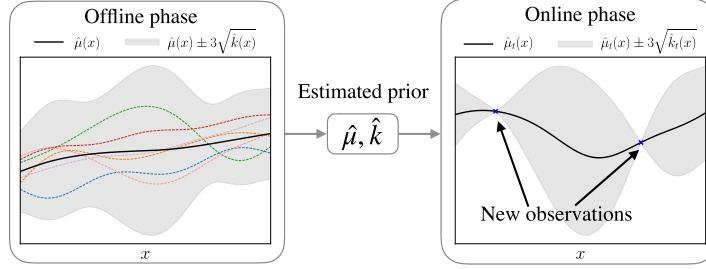

Figure 1: Our approach estimates the mean function $\hat{\mu}$ and kernel $\hat{k}$ from functions sampled from $GP(\mu, k)$ in the offline phase. Those sampled functions are illustrated by colored lines. In the online phase, a new function $f$ sampled from the same $GP(\mu, k)$ is given and we can estimate its posterior mean function $\hat{\mu}_t$ and covariance function $\hat{k}_t$ which will be used for Bayesian optimization.

Let the completed observation matrix be $Y = [\bar{y}_{ij}]_{i\in[N],j\in[M]}$. We use an unbiased sample mean and covariance estimator for $\mu$ and $k$; that is, $\hat{\mu}(\mathfrak{X}) = \frac{1}{N}Y^{\mathsf{T}}1_N$ and $\hat{k}(\mathfrak{X}) = \frac{1}{N-1}(Y - 1_N\hat{\mu}(\mathfrak{X})^{\mathsf{T}})^{\mathsf{T}}(Y - 1_N\hat{\mu}(\mathfrak{X})^{\mathsf{T}})$, where $1_N$ is an $N$ by 1 vector of ones. It is well known that $\hat{\mu}$ and $\hat{k}$ are independent and $\hat{\mu}(\mathfrak{X}) \sim \mathcal{N}(\mu(\mathfrak{X}), \frac{1}{N}(k(\mathfrak{X}) + \sigma^2\boldsymbol{I}))$, $\hat{k}(\mathfrak{X}) \sim \mathcal{W}(\frac{1}{N-1}(k(\mathfrak{X}) + \sigma^2\boldsymbol{I}), N - 1)$ [1].

**Constructing estimators of the posterior** Given noisy observations $D_t = \{(x_\tau, y_\tau)\}_{\tau=1}^{t}$, we can do Bayesian posterior inference to obtain $f \sim GP(\mu_t, k_t)$. By the GP assumption, we get

$$\mu_t(x) = \mu(x) + k(x, \boldsymbol{x}_t)(k(\boldsymbol{x}_t) + \sigma^2\boldsymbol{I})^{-1}(\boldsymbol{y}_t - \mu(\boldsymbol{x}_t)), \quad \forall x \in \mathfrak{X} \tag{1}$$

$$k_t(x, x') = k(x, x') - k(x, \boldsymbol{x}_t)(k(\boldsymbol{x}_t) + \sigma^2\boldsymbol{I})^{-1}k(\boldsymbol{x}_t, x'), \quad \forall x, x' \in \mathfrak{X}, \tag{2}$$

where $\boldsymbol{y}_t = [y_\tau]_{\tau=1}^{T}$, $\boldsymbol{x}_t = [x_\tau]_{\tau=1}^{T}$ [44]. The problem is that neither the posterior mean $\mu_t$ nor the covariance $k_t$ are computable because the Bayesian prior mean $\mu$, the kernel $k$ and the noise parameter $\sigma$ are all unknown. How to estimate $\mu_t$ and $k_t$ without knowing those prior parameters?

We introduce the following unbiased estimators for the posterior mean and covariance,

$$\hat{\mu}_t(x) = \hat{\mu}(x) + \hat{k}(x, \boldsymbol{x}_t)\hat{k}(\boldsymbol{x}_t, \boldsymbol{x}_t)^{-1}(\boldsymbol{y}_t - \hat{\mu}(\boldsymbol{x}_t)), \quad \forall x \in \mathfrak{X}, \tag{3}$$

$$\hat{k}_t(x, x') = \frac{N-1}{N-t-1}\left(\hat{k}(x, x') - \hat{k}(x, \boldsymbol{x}_t)\hat{k}(\boldsymbol{x}_t, \boldsymbol{x}_t)^{-1}\hat{k}(\boldsymbol{x}_t, x')\right), \quad \forall x, x' \in \mathfrak{X}. \tag{4}$$

Notice that unlike Eq. (1) and Eq. (2), our estimators $\hat{\mu}_t$ and $\hat{k}_t$ do not depend on any unknown values or an additional estimate of the noise parameter $\sigma$. In Lemma 1, we show that our estimators are indeed unbiased and we derive their concentration bounds.

**Lemma 1.** *Pick probability $\delta \in (0, 1)$. For any nonnegative integer $t < T$, conditioned on the observations $D_t = \{(x_\tau, y_\tau)\}_{\tau=1}^{t}$, the estimators in Eq. (3) and Eq. (4) satisfy $\mathbb{E}[\hat{\mu}_t(\mathfrak{X})] = \mu_t(\mathfrak{X}), \mathbb{E}[\hat{k}_t(\mathfrak{X})] = k_t(\mathfrak{X}) + \sigma^2\boldsymbol{I}$. Moreover, if the size of the training dataset satisfies $N \geq T + 2$, then for any input $x \in \mathfrak{X}$, with probability at least $1 - \delta$, both*

$$|\hat{\mu}_t(x) - \mu_t(x)|^2 < a_t(k_t(x) + \sigma^2) \text{ and } 1 - 2\sqrt{b_t} < \hat{k}_t(x)/(k_t(x) + \sigma^2) < 1 + 2\sqrt{b_t} + 2b_t$$

*hold, where $a_t = \frac{4\left(N-2+t+2\sqrt{t\log(4/\delta)}+2\log(4/\delta)\right)}{\delta N(N-t-2)}$ and $b_t = \frac{1}{N-t-1}\log\frac{4}{\delta}$.*

**Regret bounds** We show a near-zero upper bound on the best-sample simple regret of meta BO with GP-UCB and PI that uses specific parameter settings in Thm. 2. In particular, for both GP-UCB and PI, the regret bound converges to a residual whose scale depends on the noise level $\sigma$ in the observations.

**Theorem 2.** *Assume there exists constant $c \geq \max_{x\in\mathfrak{X}} k(x)$ and a training dataset is available whose size is $N \geq 4\log\frac{6}{\delta} + T + 2$. Then, with probability at least $1 - \delta$, the best-sample simple*

*regin in $T$ iterations of meta BO with special cases of either GP-UCB or PI satisfies*

$$r_T^{UCB} < \eta_T^{UCB}(N)\lambda_T, \ \ r_T^{PI} < \eta_T^{PI}(N)\lambda_T, \ \ \lambda_T^2 = O(\rho_T/T) + \sigma^2,$$

*where $\eta_T^{UCB}(N) = (m+C_1)(\frac{\sqrt{1+m}}{\sqrt{1-m}}+1), \eta_T^{PI}(N) = (m+C_2)(\frac{\sqrt{1+m}}{\sqrt{1-m}}+1)+C_3, m = O(\sqrt{\frac{1}{N-T}}),$ $C_1, C_2, C_3 > 0$ are constants, and $\rho_T = \max\limits_{A \in \mathfrak{X}, |A|=T} \frac{1}{2} \log |\boldsymbol{I} + \sigma^{-2}k(A)|.$*

This bound reflects how training instances $N$ and BO iterations $T$ affect the best-sample simple regret. The coefficients $\eta_T^{\text{UCB}}$ and $\eta_T^{\text{PI}}$ both converge to constants (more details in the appendix), with components converging at rate $O(1/(N-T)^{\frac{1}{2}})$. The convergence of the shared term $\lambda_T$ depends on $\rho_T$, the maximum information gain between function $f$ and up to $T$ observations $\boldsymbol{y}_T$. If, for example, each input has dimension $\mathbb{R}^d$ and $k(x, x') = x^\mathrm{T}x'$, then $\rho_T = O(d \log(T))$ [51], in which case $\lambda_T$ converges to the observational noise level $\sigma$ at rate $O(\sqrt{\frac{d \log(T)}{T}})$. Together, the bounds indicate that the best-sample simple regret of both our settings of GP-UCB and PI decreases to a constant proportional to noise level $\sigma$.

## 4.2 $\mathfrak{X} \subset \mathbb{R}^d$ is compact

For compact $\mathfrak{X} \subset \mathbb{R}^d$, we consider the primal form of GPs. We further assume that there exist basis functions $\Phi = [\phi_s]_{s=1}^K : \mathfrak{X} \to \mathbb{R}^K$, mean parameter $\boldsymbol{u} \in \mathbb{R}^K$ and covariance parameter $\Sigma \in \mathbb{R}^{K \times K}$ such that $\mu(x) = \Phi(x)^\mathrm{T}\boldsymbol{u}$ and $k(x, x') = \Phi(x)^\mathrm{T}\Sigma\Phi(x')$. Notice that $\Phi(x) \in \mathbb{R}^K$ is a column vector and $\Phi(\boldsymbol{x}_t) \in \mathbb{R}^{K \times t}$ for any $\boldsymbol{x}_t = [x_\tau]_{\tau=1}^t$. This means, for any input $x \in \mathfrak{X}$, the observation satisfies $y \sim \mathcal{N}(f(x), \sigma^2)$, where $f = \Phi(x)^\mathrm{T}W \sim GP(\mu, k)$ and the linear operator $W \sim \mathcal{N}(\boldsymbol{u}, \Sigma)$ [39]. In the following analyses, we assume the basis functions $\Phi$ are given.

We assume that a training dataset $\bar{D}_N = \{[(\bar{x}_j, \bar{y}_{ij})]_{j=1}^M\}_{i=1}^N$ is given, where $\bar{x}_j \in \mathfrak{X} \subset \mathbb{R}^d$, $y_{ij}$ are independently drawn from $\mathcal{N}(f_i(\bar{x}_j), \sigma^2)$, $f_i$ are drawn independently from $GP(\mu, k)$ and $M \geq K$.

**Estimating GP parameters** Because the basis functions $\Phi$ are given, learning the mean function $\mu$ and the kernel $k$ in the GP is equivalent to learning the mean parameter $\boldsymbol{u}$ and the covariance parameter $\Sigma$ that parameterize distribution of the linear operator $W$. Notice that $\forall i \in [N]$,

$$\bar{\boldsymbol{y}}_i = \Phi(\bar{\boldsymbol{x}})^\mathrm{T}W_i + \bar{\boldsymbol{\epsilon}}_i \sim \mathcal{N}(\Phi(\bar{\boldsymbol{x}})^\mathrm{T}\boldsymbol{u}, \Phi(\bar{\boldsymbol{x}})^\mathrm{T}\Sigma\Phi(\bar{\boldsymbol{x}}) + \sigma^2\boldsymbol{I}),$$

where $\bar{\boldsymbol{y}}_i = [\bar{y}_{ij}]_{j=1}^M \in \mathbb{R}^M$, $\bar{\boldsymbol{x}} = [\bar{x}_j]_{j=1}^M \in \mathbb{R}^{M \times d}$ and $\bar{\boldsymbol{\epsilon}}_i = [\bar{\epsilon}_{ij}]_{j=1}^M \in \mathbb{R}^M$. If the matrix $\Phi(\bar{\boldsymbol{x}}) \in \mathbb{R}^{K \times M}$ has linearly independent rows, one unbiased estimator of $W_i$ is

$$\hat{W}_i = (\Phi(\bar{\boldsymbol{x}})^\mathrm{T})^+\bar{\boldsymbol{y}}_i = (\Phi(\bar{\boldsymbol{x}})\Phi(\bar{\boldsymbol{x}})^\mathrm{T})^{-1}\Phi(\bar{\boldsymbol{x}})\bar{\boldsymbol{y}}_i \sim \mathcal{N}(\boldsymbol{u}, \Sigma + \sigma^2(\Phi(\bar{\boldsymbol{x}})\Phi(\bar{\boldsymbol{x}})^\mathrm{T})^{-1}).$$

Let $\mathsf{W} = [\hat{W}_i]_{i=1}^N \in \mathbb{R}^{N \times K}$. We use the estimator $\hat{\boldsymbol{u}} = \frac{1}{N}\mathsf{W}^\mathrm{T}1_N$ and $\hat{\Sigma} = \frac{1}{N-1}(\mathsf{W} - 1_N\hat{\boldsymbol{u}})^\mathrm{T}(\mathsf{W} - 1_N\hat{\boldsymbol{u}})$ to the estimate GP parameters. Again, $\hat{\boldsymbol{u}}$ and $\hat{\Sigma}$ are independent and $\hat{\boldsymbol{u}} \sim \mathcal{N}\left(\boldsymbol{u}, \frac{1}{N}(\Sigma + \sigma^2(\Phi(\bar{\boldsymbol{x}})\Phi(\bar{\boldsymbol{x}})^\mathrm{T})^{-1})\right), \hat{\Sigma} \sim \mathcal{W}\left(\frac{1}{N-1}\left(\Sigma + \sigma^2(\Phi(\bar{\boldsymbol{x}})\Phi(\bar{\boldsymbol{x}})^\mathrm{T})^{-1}\right), N-1\right)$ [1].

**Constructing estimators of the posterior** We assume the total number of evaluations $T < K$. Given noisy observations $D_t = \{(x_\tau, y_\tau)\}_{\tau=1}^t$, we have $\mu_t(x) = \Phi(x)^\mathrm{T}\boldsymbol{u}_t$ and $k_t(x, x') = \Phi(x)^\mathrm{T}\Sigma_t\Phi(x')$, where the posterior of $W \sim \mathcal{N}(\boldsymbol{u}_t, \Sigma_t)$ satisfies

$$\boldsymbol{u}_t = \boldsymbol{u} + \Sigma\Phi(\boldsymbol{x}_t)(\Phi(\boldsymbol{x}_t)^\mathrm{T}\Sigma\Phi(\boldsymbol{x}_t) + \sigma^2\boldsymbol{I})^{-1}(\boldsymbol{y}_t - \Phi(\boldsymbol{x}_t)^\mathrm{T}\boldsymbol{u}), \tag{5}$$

$$\Sigma_t = \Sigma - \Sigma\Phi(\boldsymbol{x}_t)(\Phi(\boldsymbol{x}_t)^\mathrm{T}\Sigma\Phi(\boldsymbol{x}_t) + \sigma^2\boldsymbol{I})^{-1}\Phi(\boldsymbol{x}_t)^\mathrm{T}\Sigma. \tag{6}$$

Similar to the strategy used in Sec. 4.1, we construct an estimator for the posterior of $W$ to be

$$\hat{\boldsymbol{u}}_t = \hat{\boldsymbol{u}} + \hat{\Sigma}\Phi(\boldsymbol{x}_t)(\Phi(\boldsymbol{x}_t)^\mathrm{T}\hat{\Sigma}\Phi(\boldsymbol{x}_t))^{-1}(\boldsymbol{y}_t - \Phi(\boldsymbol{x}_t)^\mathrm{T}\boldsymbol{u}), \tag{7}$$

$$\hat{\Sigma}_t = \frac{N-1}{N-t-1}\left(\hat{\Sigma} - \hat{\Sigma}\Phi(\boldsymbol{x}_t)(\Phi(\boldsymbol{x}_t)^\mathrm{T}\hat{\Sigma}\Phi(\boldsymbol{x}_t))^{-1}\Phi(\boldsymbol{x}_t)^\mathrm{T}\hat{\Sigma}\right). \tag{8}$$

We can compute the conditional mean and variance of the observation on $x \in \mathfrak{X}$ to be $\hat{\mu}_t(x) = \Phi(x)^\mathrm{T}\hat{\boldsymbol{u}}_t$ and $\hat{k}_t(x) = \Phi(x)^\mathrm{T}\hat{\Sigma}_t\Phi(x)$. For convenience of notation, we define $\bar{\sigma}^2(x) = \sigma^2\Phi(x)^\mathrm{T}(\Phi(\bar{\boldsymbol{x}})\Phi(\bar{\boldsymbol{x}})^\mathrm{T})^{-1}\Phi(x)$.

**Lemma 3.** *Pick probability $\delta \in (0, 1)$. Assume $\Phi(\bar{\boldsymbol{x}})$ has full row rank. For any nonnegative integer $t < T$, $T \leq K$, conditioned on the observations $D_t = \{(x_\tau, y_\tau)\}_{\tau=1}^t$, $\mathbb{E}[\hat{\mu}_t(x)] = \mu_t(x)$, $\mathbb{E}[\hat{k}_t(x)] = k_t(x) + \bar{\sigma}^2(x)$. Moreover, if the size of the training dataset satisfies $N \geq T + 2$, then for any input $x \in \mathfrak{X}$, with probability at least $1 - \delta$, both*

$$|\hat{\mu}_t(x) - \mu_t(x)|^2 < a_t(k_t(x) + \bar{\sigma}^2(x)) \text{ and } 1 - 2\sqrt{b_t} < \hat{k}_t(x)/(k_t(x) + \bar{\sigma}^2(x)) < 1 + 2\sqrt{b_t} + 2b_t$$

*hold, where $a_t = \frac{4\left(N-2+t+2\sqrt{t\log(4/\delta)}+2\log(4/\delta)\right)}{\delta N(N-t-2)}$ and $b_t = \frac{1}{N-t-1}\log\frac{4}{\delta}$.*

**Regret bounds** Similar to the finite $\mathfrak{X}$ case, we can also show a near-zero regret bound for compact $\mathfrak{X} \in \mathbb{R}^d$. The following theorem clarifies our results. The convergence rates are the same as Thm. 2. Note that $\lambda_T^2$ converges to $\bar{\sigma}^2(\cdot)$ instead of $\sigma^2$ in Thm. 2 and $\bar{\sigma}^2(\cdot)$ is proportional to $\sigma^2$ .

**Theorem 4.** *Assume all the assumptions in Thm. 2 and that $\Phi(\bar{\boldsymbol{x}})$ has full row rank. With probability at least $1 - \delta$, the best-sample simple regret in $T$ iterations of meta BO with either GP-UCB or PI satisfies*

$$r_T^{UCB} < \eta_T^{UCB}(N)\lambda_T, \ \ r_T^{PI} < \eta_T^{PI}(N)\lambda_T, \ \ \lambda_T^2 = O(\rho_T/T) + \bar{\sigma}(x_\tau)^2,$$

*where $\eta_T^{UCB}(N) = (m+C_1)(\frac{\sqrt{1+m}}{\sqrt{1-m}}+1)$, $\eta_T^{PI}(N) = (m+C_2)(\frac{\sqrt{1+m}}{\sqrt{1-m}}+1)+C_3$, $m = O(\sqrt{\frac{1}{N-T}})$, $C_1, C_2, C_3 > 0$ are constants, $\tau = \arg\min_{t\in[T]} k_{t-1}(x_t)$ and $\rho_T = \max_{A\in\mathfrak{X}, |A|=T} \frac{1}{2}\log|\boldsymbol{I}+\sigma^{-2}k(A)|$.*

### 4.3 Bounding the simple regret by the best-sample simple regret

Once we have the observations $D_T = \{(x_t, y_t)\}_{t=1}^T$, we can infer where the $\arg\max$ of the function is. For all the cases in which $\mathfrak{X}$ is discrete or compact and the acquisition function is GP-UCB or PI, we choose the inferred $\arg\max$ to be $\hat{x}_T^* = x_\tau$ where $\tau = \arg\max_{t\in[T]} y_t$. We show in Lemma 5 that with high probability, the difference between the simple regret $R_T$ and the best-sample simple regret $r_T$ is proportional to the observation noise $\sigma$.

**Lemma 5.** *With probability at least $1 - \delta$, $R_T \leq r_T + 2(2\log\frac{1}{\delta})^{\frac{1}{2}}\sigma$.*

Together with the bounds on the best-sample simple regret from Thm. 2 and Thm. 4, our result shows that, with high probability, the simple regret decreases to a constant proportional to the noise level $\sigma$ as the number of iterations and training functions increases.

## 5 Experiments

We evaluate our algorithm in four different black-box function optimization problems, involving discrete or continuous function domains. One problem is optimizing a synthetic function in $\mathbb{R}^2$, and the rest are optimizing decision variables in robotic task and motion planning problems that were used in [27][3].

At a high level, our task and motion planning benchmarks involve computing kinematically feasible collision-free motions for picking and placing objects in a scene cluttered with obstacles. This problem has a similar setup to experimental design: the robot can "experiment" by assigning values to decision variables including grasps, base poses, and object placements until it finds a feasible plan. Given the assigned values for these variables, the robot program makes

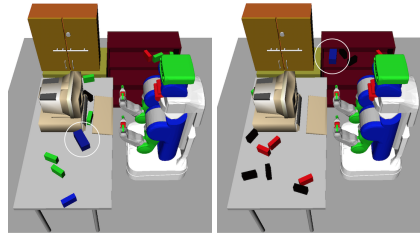

Figure 2: Two instances of a picking problem. A problem instance is defined by the arrangement and number of obstacles, which vary randomly across different instances. The objective is to select a grasp that can pick the blue box, marked with a circle, without violating kinematic and collision constraints. [27].

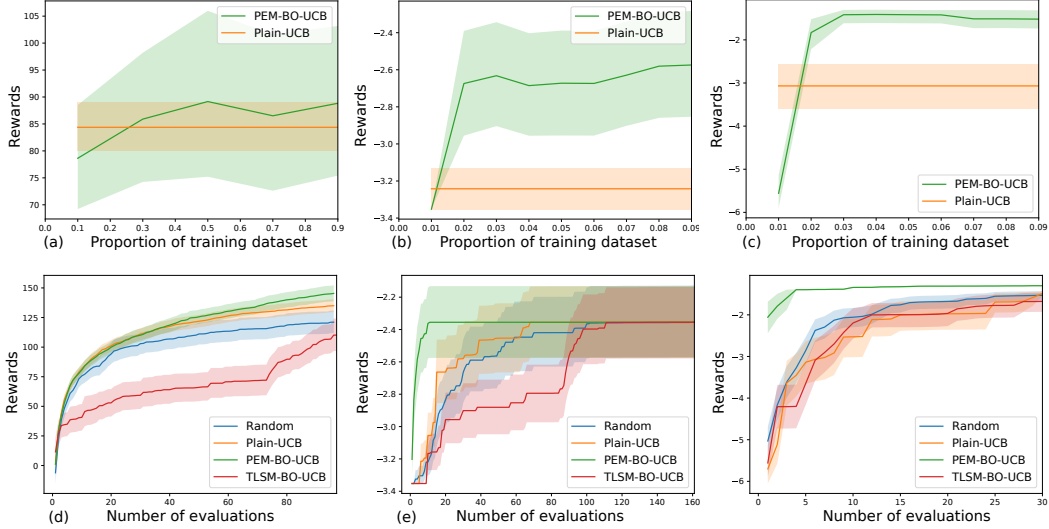

Figure 3: Learning curves (top) and rewards vs number of iterations (bottom) for optimizing synthetic functions sampled from a GP and two scoring functions from.

a call to a planner[4] which then attempts to find a sequence of motions that achieve these grasps and placements. We score the variable assignment based on the results of planning, assigning a very low score if the problem was infeasible and otherwise scoring based on plan length or obstacle clearance. An example problem is given in Figure 2.

Planning problem instances are characterized by arrangements of obstacles in the scene and the shape of the target object to be manipulated, and each problem instance defines a different score function. Our objective is to optimize the score function for a new problem instance, given sets of decision-variable and score pairs from a set of previous planning problem instances as training data.

In two robotics domains, we discretize the original function domain using samples from the past planning experience, by extracting the values of the decision variables and their scores from successful plans. This is inspired by the previous successful use of BO in a discretized domain [9] to efficiently solve an adaptive locomotion problem.

We compare our approach, called *point estimate meta Bayesian optimization* (PEM-BO), to three baseline methods. The first is a plain Bayesian optimization method that uses a kernel function to represent the covariance matrix, which we call Plain. Plain optimizes its GP hyperparameters by maximizing the data likelihood. The second is a *transfer learning sequential model-based optimization* [57] method, that, like PEM-BO, uses past function evaluations, but assumes that functions sampled from the same GP have similar response surface values. We call this method TLSM-BO. The third is random selection, which we call Random. We present the results on the UCB acquisition function in the paper and results on the PI acquisition function are available in the appendix.

In all domains, we use the $\zeta_t$ value as specified in Sec. 4. For continuous domains, we use $\Phi(x) = [\cos(x^T \beta^{(i)} + \beta_0^{(i)})]_{i=1}^K$ as our basis functions. In order to train the weights $W_i, \beta^{(i)}$, and $\beta_0^{(i)}$, we represent the function $\Phi(x)^T W_i$ with a 1-hidden-layer neural network with cosine activation function and a linear output layer with function-specific weights $W_i$. We then train this network on the entire dataset $\bar{D}_N$. Then, fixing $\Phi(x)$, for each set of pairs $(\bar{y}_i, \bar{x}_i), i = \{1 \cdots N\}$, we analytically solve the linear regression problem $y_i \approx \Phi(x_i)^T W_i$ as described in Sec. 4.2.

**Optimizing a continuous synthetic function** In this problem, the objective is to optimize a black-box function sampled from a GP, whose domain is $\mathbb{R}^2$, given a set of evaluations of different functions from the same GP. Specifically, we consider a GP with a squared exponential kernel function. The purpose of this problem is to show that PEM-BO, which estimates mean and covariance matrix based on $\bar{D}_N$, would perform similarly to BO methods that start with an appropriate prior. We have training data from $N = 100$ functions with $M = 1000$ sample points each.

Figure 3(a) shows the learning curve, when we have different portions of data. The x-axis represents the percentage of the dataset used to train the basis functions, $u$, and W from the training dataset, and the y-axis represents the best function value found after 10 evaluations on a new function. We can see that even with just ten percent of the training data points, PEM-BO performs just as well as Plain, which uses the appropriate kernel for this particular problem. Compared to PEM-BO, which can efficiently use all of the dataset, we had to limit the number of training data points for TLSM-BO to 1000, because even performing inference requires $O(NM)$ time. This leads to its noticeably worse performance than Plain and PEM-BO.

Figure 3(d) shows the how $\max_{t \in [T]} y_t$ evolves, where $T \in [1, 100]$. As we can see, PEM-BO using the UCB acquisition function performs similarly to Plain with the same acquisition function. TLSM-BO again suffers because we had to limit the number of training data points.

**Optimizing a grasp** In the robot-planning problem shown in Figure 2, the robot has to choose a grasp for picking the target object in a cluttered scene. A planning problem instance is defined by the poses of obstacles and the target objects, which changes the feasibility of a grasp across different instances.

The reward function is the negative of the length of the picking motion if the motion is feasible, and $-k \in \mathbb{R}$ otherwise, where $-k$ is a suitably lower number than the lengths of possible trajectories. We construct the discrete set of grasps by using grasps that worked in the past planning problem instances. The original space of grasps is $\mathbb{R}^{58}$, which describes position, direction, roll, and depth of a robot gripper with respect to the object, as used in [10]. For both Plain and TLSM-BO, we use squared exponential kernel function on this original grasp space to represent the covariance matrix. We note that this is a poor choice of kernel, because the grasp space includes angles, making it a non-vector space. These methods also choose a grasp from the discrete set. We train on dataset with $N = 1800$ previous problems, and let $M = 162$.

Figure 3(b) shows the learning curve with $T = 5$. The x-axis is the percentage of the dataset used for training, ranging from one percent to ten percent. Initially, when we just use one percent of the training data points, PEM-BO performs as poorly as TLSM-BO, which again, had only 1000 training data points. However, PEM-BO outperforms both TLSM-BO and Plain after that. The main reason that PEM-BO outperforms these approaches is because their prior, which is defined by the squared exponential kernel, is not suitable for this problem. PEM-BO, on the other hand, was able to avoid this problem by estimating a distribution over values at the discrete sample points that commits only to their joint normality, but not to any metric on the underlying space. These trends are also shown in Figure 3(e), where we plot $\max_{t \in [T]} y_t$ for $T \in [1, 100]$. PEM-BO outperforms the baselines significantly.

**Optimizing a grasp, base pose, and placement** We now consider a more difficult task that involves both picking and placing objects in a cluttered scene. A planning problem instance is defined by the poses of obstacles and the poses and shapes of the target object to be pick and placed. The reward function is again the negative of the length of the picking motion if the motion is feasible, and $-k \in \mathbb{R}$ otherwise. For both Plain and TLSM-BO, we use three different squared exponential kernels on the original spaces of grasp, base pose, and object placement pose respectively and then add them together to define the kernel for the whole set. For this domain, $N = 1500$, and $M = 1000$.

Figure 3(c) shows the learning curve, when $T = 5$. The x-axis is the percentage of the dataset used for training, ranging from one percent to ten percent. Initially, when we just use one percent of the training data points, PEM-BO does not perform well. Similar to the previous domain, it then significantly outperforms both TLSM-BO and Plain after increasing the training data. This is also reflected in Figure 3(f), where we plot $\max_{t \in [T]} y_t$ for $T \in [1, 100]$. PEM-BO outperforms baselines. Notice that Plain and TLSM-BO perform worse than Random, as a result of making inappropriate assumptions on the form of the kernel.

# 6 Conclusion

We proposed a new framework for meta BO that estimates its Gaussian process prior based on past experience with functions sampled from the same prior. We established regret bounds for our approach without the reliance on a known prior and showed its good performance on task and motion planning benchmark problems.

## Acknowledgments

We would like to thank Stefanie Jegelka, Tamara Broderick, Trevor Campbell, Tomás Lozano-Pérez for discussions and comments. We would like to thank Sungkyu Jung and Brian Axelrod for discussions on Wishart distributions. We gratefully acknowledge support from NSF grants 1420316, 1523767 and 1723381, from AFOSR grant FA9550-17-1-0165, from Honda Research and Draper Laboratory. Any opinions, findings, and conclusions or recommendations expressed in this material are those of the authors and do not necessarily reflect the views of our sponsors.

## Footnotes

[2]Alternatively, an upper bound $\hat{f}^*$ can be estimated adaptively [53]. Note that here we are maximizing the PI acquisition function and hence $\alpha_{t-1}^{\text{PI}}(x)$ is a negative version of what was defined in [53].

[3] Our code is available at `https://github.com/beomjoonkim/MetaLearnBO`.

[4]We use Rapidly-exploring random tree (RRT) [32] with predefined random seed, but other choices are possible.

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
