[Supplementary Material · appendix.pdf]

# Regret bounds for meta Bayesian optimization with an unknown Gaussian process prior (Appendix)

**Zi Wang**[*]
MIT CSAIL
ziw@csail.mit.edu

**Beomjoon Kim**[*]
MIT CSAIL
beomjoon@mit.edu

**Leslie Pack Kaelbling**
MIT CSAIL
lpk@csail.mit.edu

## A  Discussions and conclusions

In this section, we discuss related topics to our approach. Both theoreticians and practitioners may find this section useful in terms of clarifying theoretical insights and precautions.

### A.1  Connections and differences to empirical Bayes

In classic empirical Bayes [9, 6], we estimate the unknown parameters of the Bayesian model and usually use a point estimate to proceed any Bayesian computations. One very popular approach to estimate those unknown parameters is by maximizing the data likelihood. There also exit other variants of empirical Bayes; for example, oracle Bayes, which "shows empirical Bayes in its most frequentist mode" [4].

In this paper, we use a variant of empirical Bayes that constructs estimators for both the prior distribution and the posterior distribution. For the estimators of the posterior, we do not use a plug-in estimate like classic empirical Bayes but we construct them through Lemma. 6, which establishes the unbiasedness and concentration bounds for those estimates.

### A.2  Connections and differences to hierarchical Bayes

Hierarchical Bayes is a Bayesian hierarchical model that places priors on priors. For both of our finite $\mathfrak{X}$ case and continuous and compact $\mathfrak{X} \in \mathbb{R}^d$ case, we can write down a hierarchical Bayes model that puts a normal inverse Wishart prior on $\mu(\mathfrak{X}), k(\mathfrak{X})$ or $\boldsymbol{u}, \Sigma$.

Our approach can be viewed as a special case of the hierarchical Bayes model using point estimates to approximate the posterior. Neither our estimators nor our regret analyses depend on the prior parameters of those hierarchical Bayes models. But one may analyze the regret of BO with a better approximation from a full Bayesian perspective using hierarchical Bayes.

### A.3  Future directions

Due to the limited space, we only give the formulation of meta BO in its simple and basic settings. Our setting restricts the evaluated inputs in the training data to follow certain norms, such as where they are and how many they are, but one may certainly extend our analyses to less restrictive scenarios.

**Missing entries**   We did not consider any bounds in matrix completion [2] in our regret analyses, and proceeded with the assumption that there is no missing entry in the training data. But if missing data is a concern, one should definitely consider adapting bounds from [2] or use better estimators [8] that take into account missing entries when bounding the estimates.

---

[*]Equal contribution.

## A.4 Broader impact

We developed a statistically sound approach for meta BO with an unknown Gaussian process prior. We verified our approach on simulated task and motion planning problems. We showed that our approach is able to guide task and motion planning with good action recommendations, such that the resulting plans are better and faster to compute. We believe the theoretical guarantees may support better explanations for more practical BO approaches. In particular, our method can serve as a building block of artificial intelligence systems, and our analyses can be combined with the theoretical guarantees of other parts of the system to analyze an integrated system.

## A.5 Caveats

We did not expand the experiment sections to include applications other than task and motion planning in simulation. But there are many more scenarios that this meta BO approach will be useful. For example, our finite $\mathfrak{X}$ formulation can be used to adaptively recommend advertisements, movies or songs to Internet users, by learning a mean and kernel for those discrete items.

**Optimization objectives** Like other bandit algorithms, our approach only treats objective functions or any metrics to be optimized as *given*. Practitioners need to be very careful about what exactly they are optimizing with our approach or other optimization algorithms. For example, maximizing number of advertisement clicks or corporation profits may not be a good metric in recommendation systems; maximizing a poorly designed reward function for robotic systems may result in unexpected hazards.

**Guarantees with assumptions** In real-world applications, practitioners need to be extra cautious with our algorithm. We provided detailed assumptions and analyses, that are only based those assumptions, in Section 3 and Section 4. Outside those assumptions, we do not claim that our analyses will hold in any way. For example, in robotics applications, it may not be true that the underlying reward/cost functions are actually sampled from a GP, in which case using our method may harm the physical robot; even if those objective functions are in fact from a GP, because our regret bounds only hold with high probability, meta BO may still give dangerous actions with certain probabilities (as in frequency).

In addition, please notice that we did not provide any theoretical guarantees for using basis functions trained with neural networks. We assume those basis functions are given, which is usually not the case in practice. To the best of our knowledge, proving bounds for neural networks is very hard [5].

# B Proofs for Section 4.1

Recall that we assume $\mathfrak{X}$ is a finite set. The posterior given observations $D_t$ is $GP(\mu_t, k_t)$ where
$$\mu_t(x) = \mu(x) + k(x, \boldsymbol{x}_t)(k(\boldsymbol{x}_t) + \sigma^2 \boldsymbol{I})^{-1}(\boldsymbol{y}_t - \mu(\boldsymbol{x}_t)), \ \ \forall x \in \mathfrak{X}$$
$$k_t(x, x') = k(x, x') - k(x, \boldsymbol{x}_t)(k(\boldsymbol{x}_t) + \sigma^2 \boldsymbol{I})^{-1}k(\boldsymbol{x}_t, x'), \ \ \forall x, x' \in \mathfrak{X}.$$

We use the following estimators to approximate $\mu_t, k_t$:
$$\hat{\mu}_t(x) = \hat{\mu}(x) + \hat{k}(x, \boldsymbol{x}_t)\hat{k}(\boldsymbol{x}_t, \boldsymbol{x}_t)^{-1}(\boldsymbol{y}_t - \hat{\mu}(\boldsymbol{x}_t)), \ \ \forall x \in \mathfrak{X}, \tag{1}$$
$$\hat{k}_t(x, x') = \frac{N-1}{N-t-1}\left(\hat{k}(x, x') - \hat{k}(x, \boldsymbol{x}_t)\hat{k}(\boldsymbol{x}_t, \boldsymbol{x}_t)^{-1}\hat{k}(\boldsymbol{x}_t, x')\right), \ \ \forall x, x' \in \mathfrak{X}. \tag{2}$$

We will prove a bound on the best-sample simple regret $r_T = \max_{x \in \mathfrak{X}} f(x) - \max_{t \in [T]} f(x_t)$. The evaluated inputs $\boldsymbol{x}_t = [x_\tau]_\tau^t$ are selected either by a special case of GP-UCB using the acquisition function
$$\alpha_{t-1}^{\text{GP-UCB}}(x) = \hat{\mu}_{t-1}(x) + \zeta_t \hat{k}_{t-1}(x)^{\frac{1}{2}}, \tag{3}$$

$$\zeta_t = \frac{\left(6(N-3+t+2\sqrt{t\log\frac{6}{\delta}} + 2\log\frac{6}{\delta})/(\delta N(N-t-1))\right)^{\frac{1}{2}} + (2\log(\frac{3}{\delta}))^{\frac{1}{2}}}{(1 - 2(\frac{1}{N-t}\log\frac{6}{\delta})^{\frac{1}{2}})^{\frac{1}{2}}}, \delta \in (0,1) \tag{4}$$

or by a special case of PI using the acquisition function
$$\alpha_{t-1}^{\text{PI}}(x) = \frac{\hat{\mu}_{t-1}(x) - \hat{f}^*}{\hat{k}_{t-1}(x)^{\frac{1}{2}}}.$$

This special case of PI assumes additional information of the upper bound on function value $\hat{f}^* \geq \max_{x \in \mathfrak{X}} f(x)$.

**Corollary 1** ([11]). *Let $\delta_0 \in (0, 1)$. For any Gaussian variable $x \sim \mathcal{N}(\mu, \sigma^2), x \in \mathbb{R}$,*

$$\Pr[x - \mu \leq \zeta_0 \sigma] \geq 1 - \delta_0, \ \Pr[x - \mu \geq -\zeta_0 \sigma] \geq 1 - \delta_0$$

*where $\zeta_0 = (2 \log(\frac{1}{2\delta_0}))^{\frac{1}{2}}$.*

*Proof.* Let $z = \frac{\mu - x}{\sigma} \sim \mathcal{N}(0, 1)$. We have

$$\begin{aligned}
\Pr[z > \zeta_0] &= \int_{\zeta_0}^{+\infty} \frac{1}{\sqrt{2\pi}} e^{-z^2/2} \, \mathrm{d}z \\
&= \int_{\zeta_0}^{+\infty} \frac{1}{\sqrt{2\pi}} e^{-(z-\zeta_0)^2/2 - \zeta_0^2/2 - z\zeta_0} \, \mathrm{d}z \\
&\leq e^{-\zeta_0^2/2} \int_{\zeta_0}^{+\infty} \frac{1}{\sqrt{2\pi}} e^{-(z-\zeta_0)^2/2} \, \mathrm{d}z \\
&= \frac{1}{2} e^{-\zeta_0^2/2}.
\end{aligned}$$

Similarly, $\Pr[z < -\zeta_0] \leq \frac{1}{2} e^{-\zeta_0^2/2}$. We reach the conclusion by rearranging the constants. $\square$

**Lemma 2.** *Assume $X_1, \cdots, X_n \in \mathbb{R}^m$ are sampled i.i.d. from $\mathcal{N}(u, V)$. Suppose we estimate the sample mean to be $\hat{u} = \frac{1}{n} X^T 1_n$ and the sample covariance to be $\hat{V} = \frac{1}{n-1}(X - 1_n \hat{u}^T)^T (X - 1_n \hat{u}^T)$ where $X = [X_i]_{i=1}^n \in \mathbb{R}^{n \times m}$. Then, $\hat{u}$ and $\hat{V}$ are independent, and*

$$\hat{u} \sim \mathcal{N}(u, \frac{1}{n} V), \ \hat{V} \sim \mathcal{W}(\frac{1}{n-1} V, n-1).$$

Lemma 2 is a combination of Theorem 3.3.2 and Corollary 7.2.3 of [1]. Interested readers can find the proof of Lemma 2 in [1]. Corollary 3 directly follows Lemma 2.

**Corollary 3.** *$\hat{\mu}$ and $\hat{k}$ are independent and*

$$\hat{\mu}(\mathfrak{X}) \sim \mathcal{N}(\mu(\mathfrak{X}), \frac{1}{N}(k(\mathfrak{X}) + \sigma^2 \boldsymbol{I})), \ \hat{k}(\mathfrak{X}) \sim \mathcal{W}(\frac{1}{N-1}(k(\mathfrak{X}) + \sigma^2 \boldsymbol{I}), N-1).$$

**Corollary 4.** *For any $X \sim \mathcal{W}(v, n), v \in \mathbb{R}$ and $b > 0$, we have*

$$\Pr[\frac{X}{vn} \geq 1 + 2\sqrt{b} + 2b] \leq e^{-bn}, \ \Pr[\frac{X}{vn} \leq 1 - 2\sqrt{b}] \leq e^{-bn}.$$

*Proof.* Let $X$ be a random variable such that $X \sim \mathcal{W}(v, n)$. So $\frac{X}{v}$ is distributed according to a chi-squared distribution with $n$ degrees of freedom; namely, $\frac{X}{v} \sim \chi^2(n)$. By Lemma 1 in [7], we have

$$\Pr[\frac{X}{v} - n \geq 2\sqrt{na} + 2a] \leq e^{-a}, \ \Pr[\frac{X}{v} - n \leq -2\sqrt{na}] \leq e^{-a}.$$

As a result, if $a = bn$,

$$\Pr[\frac{X}{vn} \geq 1 + 2\sqrt{b} + 2b] \leq e^{-bn}, \ \Pr[\frac{X}{vn} \leq 1 - 2\sqrt{b}] \leq e^{-bn}.$$

$\square$

**Lemma 5.** *Let $X \in \mathbb{R}^d$ be a sample from $\mathcal{N}(w, V)$ and define $Z = (X - w)^T V^{-1}(X - w)$. Then, we have $Z \sim \chi^2(d)$. With probability at least $1 - \delta_0$, $Z < d + 2\sqrt{d \log \frac{1}{\delta_0}} + 2 \log \frac{1}{\delta_0}$.*

*Proof.* By [10], $Z \sim \chi^2(d)$. The bound on $Z$ follows Lemma 1 in [7]. $\square$

**Lemma 6.** *Pick $\delta_1 \in (0,1)$ and $\delta_2 \in (0,1)$. For any fixed non-negative integer $t < T$, conditioned on the observations $D_t = \{(x_\tau, y_\tau)\}_{\tau=1}^t$, our estimators $\hat{\mu}_t$ and $\hat{k}_t$ satisfy*

$$\mathbb{E}[\hat{\mu}_t(\mathfrak{X})] = \mu_t(\mathfrak{X}), \quad \mathbb{E}[\hat{k}_t(\mathfrak{X})] = k_t(\mathfrak{X}) + \sigma^2 I.$$

*Suppose $N \geq T + 2$. Then, for any fixed inputs $x, z \in \mathfrak{X}$,*

$$\Pr\left[\hat{\mu}_t(x) - \mu_t(x) < \iota_t\sqrt{(k_t(x) + \sigma^2)} \wedge \hat{\mu}_t(z) - \mu_t(z) > -\iota_t\sqrt{(k_t(z) + \sigma^2)}\right] \geq 1 - \delta_1, \quad (5)$$

$$\Pr[\frac{\hat{k}_t(x)}{k_t(x) + \sigma^2} < 1 + 2\sqrt{b_t} + 2b_t] \geq 1 - \delta_2, \quad \Pr[\frac{\hat{k}_t(x)}{k_t(x) + \sigma^2} > 1 - 2\sqrt{b_t}] \geq 1 - \delta_2. \quad (6)$$

*where $\iota_t = \sqrt{\frac{2\left(N-2+t+2\sqrt{t\log\frac{2}{\delta_1}}+2\log\frac{2}{\delta_1}\right)}{\delta_1 N(N-t-2)}}$ and $b_t = \frac{1}{N-t-1}\log\frac{1}{\delta_2}$.*

*Proof.* By assumption, all rows of the observation $Y = [\bar{y}_{ij}]_{i\in[N], j\in[M]}$ are sampled i.i.d. from $\mathcal{N}(\mu(\mathfrak{X}), k(\mathfrak{X}) + \sigma^2 I)$. By Corollary 3,

$$\hat{\mu}(\mathfrak{X}) \sim \mathcal{N}(\mu, \frac{1}{N}(k(\mathfrak{X}) + \sigma^2 I)), \quad \hat{k}(\mathfrak{X}) \sim \mathcal{W}(\frac{1}{N-1}(k(\mathfrak{X}) + \sigma^2 I), N-1).$$

By Proposition 8.7 in [3], we have

$$\hat{k}(x, x') - \hat{k}(x, \boldsymbol{x}_t)\hat{k}(\boldsymbol{x}_t, \boldsymbol{x}_t)^{-1}\hat{k}(\boldsymbol{x}_t, x') \sim \mathcal{W}(\frac{1}{N-1}(k_t(x, x') + \sigma^2 \mathbb{1}_{x=x'}), N-t-1).$$

Hence, the estimate $\hat{k}_t$ satisfy

$$\hat{k}_t(x) \sim \mathcal{W}(\frac{1}{N-t-1}(k_t(x) + \sigma^2), N-t-1) \quad (7)$$

Clearly, $\mathbb{E}[\hat{k}_t(x)] = k_t(x) + \sigma^2$. Now it is easy to show Eq. (6). By Corollary 4, for any fixed $t \in [T] \cup 0$ and $x, \forall \frac{1}{4} \geq b_t > 0$,

$$\Pr[\frac{\hat{k}_t(x)}{k_t(x) + \sigma^2} \geq 1 + 2\sqrt{b_t} + 2b_t] \leq e^{-b_t(N-t-1)},$$

$$\Pr[\frac{\hat{k}_t(x)}{k_t(x) + \sigma^2} \leq 1 - 2\sqrt{b_t}] \leq e^{-b_t(N-t-1)}. \quad (8)$$

where $b_t = \frac{1}{N-t-1}\log\frac{1}{\delta_2} > 0$ and $\delta_2 \in (0,1)$. Thus, we have shown Eq. (6).

We next prove the second half of the results for $\hat{\mu}_t$ in Eq. (5). We use the shorthand $S = \frac{1}{N-1}(k(\mathfrak{X}) + \sigma^2 I)$. By definition of the Wishart distributions in [3] (Definition 8.1), there exist random vectors $X_1, \cdots, X_{N-1} \in \mathbb{R}^M$ sampled iid from $\mathcal{N}(0, S), \forall i = 1, \cdots, N-1$, and $\hat{k}(\mathfrak{X}) = \sum_{i=1}^{n-1} X_i X_i^T$. We denote $X \in \mathbb{R}^{(N-1)\times M}$ as a matrix whose $i$-th row is $X_i$. Clearly, $\hat{k}(\mathfrak{X}) = X^T X$ and $\hat{k}(\mathfrak{X}_a, \mathfrak{X}_b) = X_{\cdot,a}^T X_{\cdot,b}, \forall a, b \subseteq [M]$. Let the indices of $\boldsymbol{x}_t$ in $\mathfrak{X}$ be $\Theta_t \subseteq [M]$ and the index of $x$ in $\mathfrak{X}$ be $\theta \in [M]$. Thus we have $\boldsymbol{x}_t = \mathfrak{X}_{\Theta_t}$ and $x = \mathfrak{X}_\theta$.

Conditional on $\hat{\mu}(\boldsymbol{x}_t)$ and $X_{\cdot,\Theta_t}$, the term $\hat{k}(x, \boldsymbol{x}_t)\hat{k}(\boldsymbol{x}_t)^{-1}(\boldsymbol{y}_t - \hat{\mu}(\boldsymbol{x}_t))$ is a weighted sum of independent Gaussian variables, because $X_{\cdot,\theta}^T$ consists of independent Gaussian variables and $\hat{k}(x, \boldsymbol{x}_t)\hat{k}(\boldsymbol{x}_t)^{-1}(\boldsymbol{y}_t - \hat{\mu}(\boldsymbol{x}_t)) = X_{\cdot,\theta}^T P$ where $P = X_{\cdot,\Theta_t}\left(X_{\cdot,\Theta_t}^T X_{\cdot,\Theta_t}\right)^{-1}(\boldsymbol{y}_t - \hat{\mu}(\boldsymbol{x}_t))$. Recall that $X_i \sim \mathcal{N}(0, S)$; hence, we have

$$X_{\cdot,\theta} \mid X_{\cdot,\Theta_t} \sim \mathcal{N}(X_{\cdot,\Theta_t} S_{\Theta_t}^{-1} S_{\Theta_t,\theta}, I_{N-1} \otimes S_{\theta|\Theta_t}),$$

where $S_{\theta|\Theta_t} = S_\theta - S_{\theta,\Theta_t} S_{\Theta_t}^{-1} S_{\theta,\Theta_t}^T$. As a result, the Gaussian variable $X_{\cdot,\theta}^T P$ has mean

$$\mathbb{E}[X_{\cdot,\theta}^T P \mid \hat{\mu}(\boldsymbol{x}_t), X_{\cdot,\Theta_t}] = S_{\theta,\Theta_t} S_{\Theta_t}^{-1}(\boldsymbol{y}_t - \hat{\mu}(\boldsymbol{x}_t))$$

and variance

$$\mathbb{V}[X_{\cdot,\theta}^T P \mid \hat{\mu}(\boldsymbol{x}_t), X_{\cdot,\Theta_t}] = (\boldsymbol{y}_t - \hat{\mu}(\boldsymbol{x}_t))^T \hat{k}(\boldsymbol{x}_t)^{-1}(\boldsymbol{y}_t - \hat{\mu}(\boldsymbol{x}_t))S_{\theta|\Theta_t}.$$

By independence between $\hat{k}(\mathfrak{X})$ and $\hat{\mu}(\mathfrak{X})$ shown in Corollary 3, we can show that $\hat{k}(x, \boldsymbol{x}_t)$ and $\hat{\mu}(x)$ are independent conditional on $\hat{\mu}(\boldsymbol{x}_t)$ and $\hat{k}(\boldsymbol{x}_t)$, by noting that

$$p(\hat{\mu}(\mathfrak{X}), \hat{k}(\mathfrak{X})) = p(\hat{\mu}(\mathfrak{X}))p(\hat{k}(\mathfrak{X}))$$
$$\Rightarrow p(\hat{\mu}(\boldsymbol{x}_t \cup \{x\}), \hat{k}(\boldsymbol{x}_t \cup \{x\})) = p(\hat{\mu}(\boldsymbol{x}_t \cup \{x\}))p(\hat{k}(\boldsymbol{x}_t \cup \{x\}))$$
$$\Rightarrow p(\hat{\mu}(\boldsymbol{x}_t \cup \{x\}), \hat{k}(\boldsymbol{x}_t \cup \{x\})) = p(\hat{\mu}(\boldsymbol{x}_t \cup \{x\}) \mid \hat{k}(\boldsymbol{x}_t))p(\hat{k}(\boldsymbol{x}_t \cup \{x\}) \mid \hat{\mu}(\boldsymbol{x}_t))$$
$$\Rightarrow p(\hat{\mu}(x), \hat{k}(x), \hat{k}(x, \boldsymbol{x}_t) \mid \hat{\mu}(\boldsymbol{x}_t), \hat{k}(\boldsymbol{x}_t)) = p(\hat{\mu}(x) \mid \hat{\mu}(\boldsymbol{x}_t), \hat{k}(\boldsymbol{x}_t))p(\hat{k}(x), \hat{k}(x, \boldsymbol{x}_t) \mid \hat{\mu}(\boldsymbol{x}_t), \hat{k}(\boldsymbol{x}_t))$$
$$\Rightarrow p(\hat{\mu}(x), \hat{k}(x, \boldsymbol{x}_t) \mid \hat{\mu}(\boldsymbol{x}_t), \hat{k}(\boldsymbol{x}_t)) = p(\hat{\mu}(x) \mid \hat{\mu}(\boldsymbol{x}_t), \hat{k}(\boldsymbol{x}_t)))p(\hat{k}(x, \boldsymbol{x}_t) \mid \hat{\mu}(\boldsymbol{x}_t)), \hat{k}(\boldsymbol{x}_t)).$$

Hence, $\hat{\mu}(x)$ and $X_{\cdot, \theta}^{\mathrm{T}} P = \hat{k}(x, \boldsymbol{x}_t)\hat{k}(\boldsymbol{x}_t)^{-1}(\boldsymbol{y}_t - \hat{\mu}(\boldsymbol{x}_t))$ are independent conditional on $\hat{\mu}(\boldsymbol{x}_t)$ and $\hat{k}(\boldsymbol{x}_t)$. Moreover, $X_{\cdot, \theta}^{\mathrm{T}} P$ is dependent on $X_{\cdot, \Theta_t}$ only through $\hat{k}(\boldsymbol{x}_t) = X_{\cdot, \Theta_t}^{\mathrm{T}} X_{\cdot, \Theta_t}$; hence, we have

$$\hat{\mu}_t(x) \mid \hat{\mu}(\boldsymbol{x}_t), \hat{k}(\boldsymbol{x}_t) \sim \mathcal{N}(\bar{\mu}, \bar{S}), \tag{9}$$

By linearity of expectation and the Bienaymé formula,

$$\bar{\mu} = \mathbb{E}[\hat{\mu}(x) \mid \hat{\mu}(\boldsymbol{x}_t)] + k(x, \boldsymbol{x}_t)(k(\boldsymbol{x}_t) + \sigma^2 \boldsymbol{I})^{-1}(\boldsymbol{y}_t - \hat{\mu}(\boldsymbol{x}_t)) \tag{10}$$
$$= \mu(x) + k(x, \boldsymbol{x}_t)(k(\boldsymbol{x}_t) + \sigma^2 \boldsymbol{I})^{-1}(\boldsymbol{y}_t - \mu(\boldsymbol{x}_t))$$
$$= \mu_t(x),$$
$$\bar{S} = \mathbb{V}[\hat{\mu}(x) \mid \hat{\mu}(\boldsymbol{x}_t)] + \frac{(\boldsymbol{y}_t - \hat{\mu}(\boldsymbol{x}_t))^{\mathrm{T}} \hat{k}(\boldsymbol{x}_t)^{-1}(\boldsymbol{y}_t - \hat{\mu}(\boldsymbol{x}_t))(k_t(x) + \sigma^2)}{n-1}, \tag{11}$$
$$= \frac{k_t(x) + \sigma^2}{N} + \frac{(\boldsymbol{y}_t - \hat{\mu}(\boldsymbol{x}_t))^{\mathrm{T}} \hat{k}(\boldsymbol{x}_t)^{-1}(\boldsymbol{y}_t - \hat{\mu}(\boldsymbol{x}_t))(k_t(x) + \sigma^2)}{N-1}.$$

In Eq. (10) and Eq. (11), we use the conditional Gaussian distribution for $\hat{\mu}(x)$ as follows

$$\hat{\mu}(x) \mid \hat{\mu}(\boldsymbol{x}_t) \sim \mathcal{N}(\mu(x) + k(x, \boldsymbol{x}_t)(k(\boldsymbol{x}_t) + \sigma^2 \boldsymbol{I})^{-1}(\hat{\mu}(\boldsymbol{x}_t) - \mu(\boldsymbol{x}_t)), \frac{k_t(x) + \sigma^2}{N}).$$

By the law of total expectation,

$$\mathbb{E}[\hat{\mu}_t(x)] = \mathbb{E}\left[\mathbb{E}[\hat{\mu}_t(x) \mid \hat{\mu}(\boldsymbol{x}_t), \hat{k}(\boldsymbol{x}_t)]\right] = \mu_t(x). \tag{12}$$

By the law of total variance,

$$\mathbb{V}[\hat{\mu}_t(x)] = \mathbb{E}\left[\mathbb{V}[\hat{\mu}_t(x) \mid \hat{\mu}(\boldsymbol{x}_t), \hat{k}(\boldsymbol{x}_t)]\right] + \mathbb{V}\left[\mathbb{E}[\hat{\mu}_t(x) \mid \hat{\mu}(\boldsymbol{x}_t), \hat{k}(\boldsymbol{x}_t)]\right]$$
$$= \mathbb{E}\left[\bar{S}\right] + \mathbb{V}\left[\bar{\mu}\right]$$
$$= \frac{\left(N - 2 + (\boldsymbol{y}_t - \mu(\boldsymbol{x}_t))^{\mathrm{T}}(k(\boldsymbol{x}_t) + \sigma^2 \boldsymbol{I})^{-1}(\boldsymbol{y}_t - \mu(\boldsymbol{x}_t))\right)(k_t(x) + \sigma^2)}{N(N - t - 2)}$$
$$= \frac{(N - 2 + K_{\boldsymbol{x}_t, \boldsymbol{y}_t})(k_t(x) + \sigma^2)}{N(N - t - 2)}.$$

where $K_{\boldsymbol{x}_t, \boldsymbol{y}_t} = (\boldsymbol{y}_t - \mu(\boldsymbol{x}_t))^{\mathrm{T}}(k(\boldsymbol{x}_t) + \sigma^2 \boldsymbol{I})^{-1}(\boldsymbol{y}_t - \mu(\boldsymbol{x}_t))$.

Notice that $\hat{\mu}_t(x) \mid \hat{\mu}(\boldsymbol{x}_t), \hat{k}(\boldsymbol{x}_t)$ in Eq. (9) is a normal distribution centered at $\mu_t(x)$ regardless of the conditional distribution. So the distribution of $\hat{\mu}_t(x)$ must be symmetric with a center at $\mu_t(x)$. Hence, applying Chebyshev's inequality, we have

$$\Pr\left[\hat{\mu}_t(x) - \mu_t(x) < \sqrt{\frac{(N - 2 + K_{\boldsymbol{x}_t, \boldsymbol{y}_t})(k_t(x) + \sigma^2)}{2\delta_1' N(N - t - 2)}}\right] \geq 1 - \delta_1',$$

$$\Pr\left[\hat{\mu}_t(x) - \mu_t(x) > -\sqrt{\frac{(N - 2 + K_{\boldsymbol{x}_t, \boldsymbol{y}_t})(k_t(x) + \sigma^2)}{2\delta_1' N(N - t - 2)}}\right] \geq 1 - \delta_1'.$$

Notice that the randomness of $K_{\boldsymbol{x}_t, \boldsymbol{y}_t}$ is from $\boldsymbol{y}_t$ and $\boldsymbol{y}_t \sim \mathcal{N}(\mu(\boldsymbol{x}_t), k(\boldsymbol{x}_t) + \sigma^2 \boldsymbol{I})$. So we can further bound $K_{\boldsymbol{x}_t, \boldsymbol{y}_t} \leq t + 2\sqrt{t \log \frac{1}{\delta_1''}} + 2 \log \frac{1}{\delta_1''}$ with probability at most $\delta_1''$ by Corollary 4. Hence, if we set $\delta_1' = \frac{\delta_1}{4}$ and $\delta_1'' = \frac{\delta_1}{2}$, with probability at least $1 - \delta_1$, we have

$$\hat{\mu}_t(x) - \mu_t(x) < \iota_t \sqrt{(k_t(x) + \sigma^2)} \wedge \hat{\mu}_t(z) - \mu_t(z) > -\iota_t \sqrt{(k_t(z) + \sigma^2)},$$

for fixed inputs $x, x'$.

Combining this result and the results in Eq. (7), Eq. (8), Eq. (12), we proved the lemma. $\qquad\square$

**Lemma 7** (Lemma 1 in the paper). *Pick probability $\delta \in (0, 1)$. For any nonnegative integer $t < T$, conditioned on the observations $D_t = \{(x_\tau, y_\tau)\}_{\tau=1}^t$, the estimators in Eq. (1) and Eq. (2) satisfy $\mathbb{E}[\hat{\mu}_t(\mathfrak{X})] = \mu_t(\mathfrak{X}), \mathbb{E}[\hat{k}_t(\mathfrak{X})] = k_t(\mathfrak{X}) + \sigma^2 \boldsymbol{I}$. Moreover, if the size of the training dataset satisfy $N \geq T + 2$, then for any input $x \in \mathfrak{X}$, with probability at least $1 - \delta$, both*

$$|\hat{\mu}_t(x) - \mu_t(x)|^2 < a_t(k_t(x) + \sigma^2) \ \text{and} \ 1 - 2\sqrt{b_t} < \hat{k}_t(x)/(k_t(x) + \sigma^2) < 1 + 2\sqrt{b_t} + 2b_t$$

*hold, where $a_t = \frac{4\left(N - 2 + t + 2\sqrt{t \log(4/\delta)} + 2\log(4/\delta)\right)}{\delta N (N - t - 2)}$ and $b_t = \frac{1}{N - t - 1} \log \frac{4}{\delta}$.*

*Proof.* By a union bound on Eq. (8) of Lemma 6, we have

$$\Pr\left[1 - 2\sqrt{b_t} < \hat{k}_t(x)/(k_t(x) + \sigma^2) < 1 + 2\sqrt{b_t} + 2b_t\right] \geq 1 - 2e^{-b_t(N - t - 1)}$$

where $b_t = \frac{1}{N - t - 1} \log \frac{1}{\delta_2} > 0$ and $\delta_2 \in (0, 1)$. By Lemma 6, we also have

$$\Pr\left[\hat{\mu}_t(x) - \mu_t(x) < \iota_t \sqrt{(k_t(x) + \sigma^2)} \wedge \hat{\mu}_t(z) - \mu_t(z) > -\iota_t \sqrt{(k_t(z) + \sigma^2)}\right] \geq 1 - \delta_1,$$

where $\iota_t = \sqrt{\frac{2\left(N - 2 + t + 2\sqrt{t \log \frac{2}{\delta_1}} + 2\log \frac{2}{\delta_1}\right)}{\delta_1 N (N - t - 2)}}$. We get the conclusion of this lemma by setting $a_t = \iota_t, \delta_1 = \delta_2 = \frac{\delta}{2}$, and $z = x$. $\qquad\square$

**Corollary 8** (Corollary of Bernoulli's inequality). *For any $0 \leq x \leq c$ and $a > 0$, we have $x \leq \frac{c \log(1 + \frac{ax}{c})}{\log(1 + a)}$.*

*Proof.* By Bernoulli's inequality, $(1 + a)^{\frac{x}{c}} \leq 1 + \frac{ax}{c}$. Because $\log(1 + a) > 0$, by rearranging, we have $x \leq \frac{c \log(1 + \frac{ax}{c})}{\log(1 + a)}$. $\qquad\square$

**Lemma 9.** *For any $0 \leq x \leq c$ and $a > 0$, we have $\sqrt{x} < \sqrt{x + a} - \frac{a}{2\sqrt{c + a}}$.*

*Proof.* Numerically, for any $n \geq 1$, $\frac{1}{\sqrt{n}} < 2\sqrt{n} - 2\sqrt{n - 1}$ [12]. Let $n = \frac{x}{a} + 1$. Then, we have

$$\frac{1}{\sqrt{\frac{x}{a} + 1}} < 2\sqrt{\frac{x}{a} + 1} - 2\sqrt{\frac{x}{a}}$$

$$\frac{a}{\sqrt{a + c}} < \frac{a}{\sqrt{a + x}} < 2\sqrt{x + a} - 2\sqrt{x}$$

$$\sqrt{x} < \sqrt{x + a} - \frac{a}{2\sqrt{a + c}}. \qquad\square$$

**Lemma 10** (Lemma 5.3 of [11]). *Let $\boldsymbol{x}_T = [x_t]_{t=1}^T \subseteq \mathfrak{X}$. The mutual information between the function values $f(\boldsymbol{x}_T)$ and their observations $\boldsymbol{y}_T = [y_t]_{t=1}^T$ satisfy*

$$I(f(\boldsymbol{x}_T); \boldsymbol{y}_T) = \frac{1}{2} \log \det(\boldsymbol{I} + \sigma^{-2} k(\boldsymbol{x}_t)) = \frac{1}{2} \sum_{t=1}^k \log(1 + \sigma^{-2} k_{t-1}(x_t)).$$

**Theorem 11.** *Assume there exist constant $c \geq \max_{x \in \mathfrak{X}} k(x)$ and a training dataset is available whose size is $N \geq 4 \log \frac{6}{\delta} + T + 2$. Define*

$$\iota_{t-1} = \sqrt{\frac{6 \left( N - 3 + t + 2\sqrt{t \log \frac{6}{\delta}} + 2 \log \frac{6}{\delta} \right)}{\delta N (N - t - 1)}}, \quad b_{t-1} = \frac{1}{N-t} \log \frac{6}{\delta}, \quad \textit{for any } t \in [T],$$

*and $\rho_T = \max_{A \in \mathfrak{X}, |A| = T} \frac{1}{2} \log |\boldsymbol{I} + \sigma^{-2} k(A)|$. Then, with probability at least $1 - \delta$, the best-sample simple regret in $T$ iterations of meta BO with GP-UCB that uses Eq. (4) as its hyperparameter satisfies*

$$r_T^{GP\text{-}UCB} \leq \eta^{GP\text{-}UCB} \sqrt{\frac{2c\rho_T}{T \log(1 + c\sigma^{-2})} + \sigma^2} - \frac{(2 \log(\frac{3}{\delta}))^{\frac{1}{2}} \sigma^2}{\sqrt{c + \sigma^2}},$$

*where $\eta^{GP\text{-}UCB} = \left( \frac{\iota_{T-1} + (2 \log(\frac{3}{\delta}))^{\frac{1}{2}}}{\sqrt{1 - 2\sqrt{b_{T-1}}}} \sqrt{1 + 2\sqrt{b_{T-1}} + 2b_{T-1}} + \iota_{T-1} + (2 \log(\frac{3}{\delta}))^{\frac{1}{2}} \right)$.*

*With probability at least $1 - \delta$, the best-sample simple regret in $T$ iterations of meta BO with PI that uses $\hat{f}^* \geq \max_{x \in \mathfrak{X}} f(x)$ as its target value satisfies*

$$r_T^{PI} < \eta^{PI} \sqrt{\frac{2c\rho_T}{T \log(1 + c\sigma^{-2})} + \sigma^2} - \frac{(2 \log(\frac{3}{2\delta}))^{\frac{1}{2}} \sigma^2}{2\sqrt{c + \sigma^2}},$$

*where $\eta^{PI} = \left( \frac{\hat{f}^* - \mu_{\tau-1}(x_*)}{\sqrt{k_{\tau-1}(x_*) + \sigma^2}} + \iota_{\tau-1} \right) \sqrt{\frac{1 + 2b_{\tau-1}^{\frac{1}{2}} + 2b_{\tau-1}}{1 - 2b_{\tau-1}^{\frac{1}{2}}}} + \iota_{\tau-1} + (2 \log(\frac{3}{2\delta}))^{\frac{1}{2}}, \quad \tau = \arg\min_{t \in [T]} k_{t-1}(x_t)$.*

*Proof.* We first show the regret bound for GP-UCB with our estimators of prior and posterior. All of the probabilities mentioned in the proofs need to be interpreted in a frequentist manner. Let $\tau = \arg\min_{t \in [T]} k_{t-1}(x_t)$. By Corollary 1, with probability at least $1 - \frac{\delta}{3}$,

$$
\begin{aligned}
r_T^{\text{GP-UCB}} &= f^* - \max_{t \in [T]} f(x_t) \\
&\leq f^* - f(x_\tau) \\
&\leq f^* - \mu_{\tau-1}(x_\tau) + \mu_{\tau-1}(x_\tau) - f(x_\tau) \\
&\leq \mu_{\tau-1}(x_*) + \zeta' \sqrt{k_{\tau-1}(x_*)} - \mu_{\tau-1}(x_\tau) + \zeta' \sqrt{k_{\tau-1}(x_\tau)},
\end{aligned}
$$

where $\zeta' = (2 \log(\frac{3}{\delta}))^{\frac{1}{2}}$. By Lemma 6, with probability at least $1 - \frac{\delta}{3}$,

$$\mu_{\tau-1}(x_*) - \mu_{\tau-1}(x_\tau) < \hat{\mu}_{\tau-1}(x_*) - \hat{\mu}_{\tau-1}(x_\tau) + \iota_{\tau-1}\sqrt{k_{\tau-1}(x_*) + \sigma^2} + \iota_{\tau-1}\sqrt{k_{\tau-1}(x_\tau) + \sigma^2},$$

where $\iota_t = \sqrt{\frac{6 \left( N - 2 + t + 2\sqrt{t \log \frac{6}{\delta}} + 2 \log \frac{6}{\delta} \right)}{\delta N (N - t - 2)}} \leq \iota_{T-1}$.

Lemma 6 and Lemma 9 also show that with probability at least $1 - \frac{\delta}{6}$, we have

$$\sqrt{k_{\tau-1}(x_*)} \leq \sqrt{k_{\tau-1}(x_*) + \sigma^2} - \frac{\sigma^2}{2\sqrt{c + \sigma^2}} < \sqrt{\frac{\hat{k}_{\tau-1}(x_*)}{1 - 2\sqrt{b_{\tau-1}}}} - \frac{\sigma^2}{2\sqrt{c + \sigma^2}}$$

where $b_t = \frac{1}{N-t-1} \log \frac{6}{\delta} \leq b_{T-1} \in (0, \frac{1}{4})$. Notice that because of the input selection strategy of GP-UCB with $\zeta_t = \frac{\iota_{t-1} + \zeta'}{\sqrt{1 - 2\sqrt{b_{t-1}}}}$, the following inequality holds with probability at least $1 - \frac{\delta}{6}$,

$$\hat{\mu}_{\tau-1}(x_*) + (\iota_{t-1} + \zeta')\sqrt{k_{\tau-1}(x_*) + \sigma^2} \leq \hat{\mu}_{\tau-1}(x_*) + \zeta_t\sqrt{\hat{k}_{\tau-1}(x_*)}$$

$$\leq \hat{\mu}_{\tau-1}(x_\tau) + \zeta_t\sqrt{\hat{k}_{\tau-1}(x_\tau)}.$$

Hence, with probability at least $1 - \delta$,

$$r_T^{\text{GP-UCB}} \leq \mu_{\tau-1}(x_*) + \zeta'\sqrt{k_{\tau-1}(x_*) + \sigma^2} - \mu_{\tau-1}(x_\tau) + \zeta'\sqrt{k_{\tau-1}(x_\tau) + \sigma^2} - \frac{\zeta'\sigma^2}{\sqrt{c + \sigma^2}}$$

$$< \hat{\mu}_{\tau-1}(x_*) - \hat{\mu}_{\tau-1}(x_\tau) + (\iota_{t-1} + \zeta')(\sqrt{k_{\tau-1}(x_*) + \sigma^2} + \sqrt{k_{\tau-1}(x_\tau) + \sigma^2}) - \frac{\zeta'\sigma^2}{\sqrt{c + \sigma^2}}$$

$$\leq \zeta_t\sqrt{\hat{k}_{\tau-1}(x_\tau)} + (\iota_{t-1} + \zeta')\sqrt{k_{\tau-1}(x_\tau) + \sigma^2} - \frac{\zeta'\sigma^2}{\sqrt{c + \sigma^2}}$$

$$< (\zeta_t\sqrt{1 + 2\sqrt{b_{t-1}} + 2b_{t-1}} + \iota_{t-1} + \zeta')\sqrt{k_{\tau-1}(x_\tau) + \sigma^2} - \frac{\zeta'\sigma^2}{\sqrt{c + \sigma^2}}$$

$$< \eta^{\text{GP-UCB}}\sqrt{k_{\tau-1}(x_\tau) + \sigma^2} - \frac{\zeta'\sigma^2}{\sqrt{c + \sigma^2}},$$

where $\eta^{\text{GP-UCB}} = (\frac{\iota_{T-1} + \zeta'}{\sqrt{1 - 2\sqrt{b_{T-1}}}}\sqrt{1 + 2\sqrt{b_{T-1}} + 2b_{T-1}} + \iota_{T-1} + \zeta')$. By Corollary 8 and the fact that $\tau = \arg\min_{t \in [T]} k_{t-1}(x_t)$, we have

$$k_{\tau-1}(x_\tau) \leq \frac{1}{T}\sum_{t=1}^{T} k_{t-1}(x_t)$$

$$\leq \frac{1}{T}\sum_{t=1}^{T} \frac{c\log(1 + \frac{c\sigma^{-2}k_{t-1}(x_t)}{c})}{\log(1 + c\sigma^{-2})}$$

$$= \frac{c}{T\log(1 + c\sigma^{-2})}\sum_{t=1}^{T} \log(1 + \sigma^{-2}k_{t-1}(x_t)).$$

Notice that here Corollary 8 applies because $0 \leq k_{\tau-1}(x_\tau) \leq c$.

By Lemma 10, $I(f(\boldsymbol{x}_T); \boldsymbol{y}_T) = \frac{1}{2}\sum_{t=1}^{T}\log(1 + \sigma^{-2}k_{t-1}(x_t)) \leq \rho_T$, so

$$k_{\tau-1}(x_\tau) \leq \frac{2c\rho_T}{T\log(1 + c\sigma^{-2})},$$

which implies

$$r_T^{\text{GP-UCB}} < \eta\sqrt{\frac{2c\rho_T}{T\log(1 + c\sigma^{-2})} + \sigma^2} - \frac{\zeta'\sigma^2}{\sqrt{c + \sigma^2}}.$$

Next, we show the proof for a special case of PI with $\hat{f}^*$, an upper bound on $f$, as its target value. Again, by Corollary 1, with probability at least $1 - \frac{\delta}{3}$,

$$r_T^{\text{PI}} = \hat{f}^* - \max_{t \in [T]} f(x_t)$$

$$\leq \hat{f}^* - f(x_\tau)$$

$$\leq \hat{f}^* - \mu_{\tau-1}(x_\tau) + \mu_{\tau-1}(x_\tau) - f(x_\tau)$$

$$\leq \hat{f}^* - \mu_{\tau-1}(x_\tau) + \zeta'\sqrt{k_{\tau-1}(x_\tau)},$$

where $\zeta' = (2\log(\frac{3}{2\delta}))^{\frac{1}{2}}$ and $\tau = \arg\min_{t \in [T]} k_{t-1}(x_t)$. By Lemma 6 and the selection strategy of PI, with probability at least $1 - \frac{2\delta}{3}$,

$$\hat{f}^* - \mu_{\tau-1}(x_\tau) < \hat{f}^* - \hat{\mu}_{\tau-1}(x_\tau) + \iota_{\tau-1}\sqrt{k_{\tau-1}(x_\tau) + \sigma^2}$$

$$\leq \frac{\hat{f}^* - \hat{\mu}_{\tau-1}(x_*)}{\sqrt{\hat{k}_{\tau-1}(x_*)}}\sqrt{\hat{k}_{\tau-1}(x_\tau)} + \iota_{\tau-1}\sqrt{k_{\tau-1}(x_\tau) + \sigma^2}$$

$$\leq \frac{\hat{f}^* - \mu_{\tau-1}(x_*) + \iota_{\tau-1}\sqrt{k_{\tau-1}(x_*) + \sigma^2}}{\sqrt{\hat{k}_{\tau-1}(x_*)}}\sqrt{\hat{k}_{\tau-1}(x_\tau)} + \iota_{\tau-1}\sqrt{k_{\tau-1}(x_\tau) + \sigma^2}$$

$$< \left((\frac{\hat{f}^* - \mu_{\tau-1}(x_*)}{\sqrt{k_{\tau-1}(x_*) + \sigma^2}} + \iota_{\tau-1})\sqrt{\frac{1 + 2b_{\tau-1}^{\frac{1}{2}} + 2b_{\tau-1}}{1 - 2b_{\tau-1}^{\frac{1}{2}}}} + \iota_{\tau-1}\right)\sqrt{k_{\tau-1}(x_\tau) + \sigma^2}.$$

Hence, with probability at least $1 - \delta$, the best-sample simple regret of PI satisfy

$$r_T^{\text{PI}} < \eta^{\text{PI}}\sqrt{\frac{2c\rho_T}{T\log(1 + c\sigma^{-2})} + \sigma^2} - \frac{\zeta'\sigma^2}{2\sqrt{c + \sigma^2}},$$

where $\eta^{\text{PI}} = (\frac{\hat{f}^* - \mu_{\tau-1}(x_*)}{\sqrt{k_{\tau-1}(x_*) + \sigma^2}} + \iota_{\tau-1})\sqrt{\frac{1 + 2b_{\tau-1}^{\frac{1}{2}} + 2b_{\tau-1}}{1 - 2b_{\tau-1}^{\frac{1}{2}}}} + \iota_{\tau-1} + \zeta'.$ $\qquad\square$

**Theorem 12** (Theorem 2 in th paper). *Assume there exist constant $c \geq \max_{x \in \mathfrak{X}} k(x)$ and a training dataset is available whose size is $N \geq 4\log\frac{6}{\delta} + T + 2$. Then, with probability at least $1 - \delta$, the best-sample simple regret in $T$ iterations of meta BO with special cases of either GP-UCB or PI satisfies*

$$r_T^{UCB} < \eta_T^{UCB}(N)\lambda_T, \quad r_T^{PI} < \eta_T^{PI}(N)\lambda_T, \quad \lambda_T^2 = O(\rho_T/T) + \sigma^2,$$

*where $\eta_T^{UCB}(N) = (m + C_1)(\frac{\sqrt{1+m}}{\sqrt{1-m}} + 1)$, $\eta_T^{PI}(N) = (m + C_2)(\frac{\sqrt{1+m}}{\sqrt{1-m}} + 1) + C_3$, $m = O(\sqrt{\frac{1}{N-T}})$, $C_1, C_2, C_3 > 0$ are constants, and $\rho_T = \max_{A \in \mathfrak{X}, |A| = T} \frac{1}{2}\log|\mathbf{I} + \sigma^{-2}k(A)|.$*

*Proof.* This theorem is a condensed version of Thm. 11 with big O notations. $\qquad\square$

## C  Proofs for Section 4.2

Recall that we assume $\mathfrak{X}$ is a compact set which is a subset of $\mathbb{R}^d$. We only considers a special case of GPs that assumes $f(x) = \Phi(x)^\mathsf{T}W$, $W \sim \mathcal{N}(\mathbf{u}, \Sigma)$ and the basis functions $\Phi(x) \in \mathbb{R}^K$ are given. The mean function and kernel are defined as

$$\mu(x) = \Phi(x)^\mathsf{T}\mathbf{u} \quad \text{and} \quad k(x) = \Phi(x)^\mathsf{T}\Sigma\Phi(x).$$

Given noisy observations $D_t = \{(x_\tau, y_\tau)\}_{\tau=1}^t, t \leq K$, we have

$$\mu_t(x) = \Phi(x)^\mathsf{T}\mathbf{u}_t \quad \text{and} \quad k_t(x, x') = \Phi(x)^\mathsf{T}\Sigma_t\Phi(x'),$$

where the posterior of $W \sim \mathcal{N}(\mathbf{u}_t, \Sigma_t)$ satisfies

$$\mathbf{u}_t = \mathbf{u} + \Sigma\Phi(\mathbf{x}_t)(\Phi(\mathbf{x}_t)^\mathsf{T}\Sigma\Phi(\mathbf{x}_t) + \sigma^2\mathbf{I})^{-1}(\mathbf{y}_t - \Phi(\mathbf{x}_t)^\mathsf{T}\mathbf{u}),$$
$$\Sigma_t = \Sigma - \Sigma\Phi(\mathbf{x}_t)(\Phi(\mathbf{x}_t)^\mathsf{T}\Sigma\Phi(\mathbf{x}_t) + \sigma^2\mathbf{I})^{-1}\Phi(\mathbf{x}_t)^\mathsf{T}\Sigma.$$

Our estimators for $\mathbf{u}_t$ and $\Sigma_t$ are

$$\hat{\mathbf{u}}_t = \hat{\mathbf{u}} + \hat{\Sigma}\Phi(\mathbf{x}_t)(\Phi(\mathbf{x}_t)^\mathsf{T}\hat{\Sigma}\Phi(\mathbf{x}_t))^{-1}(\mathbf{y}_t - \Phi(\mathbf{x}_t)^\mathsf{T}\mathbf{u}),$$
$$\hat{\Sigma}_t = \frac{N-1}{N-t-1}\left(\hat{\Sigma} - \hat{\Sigma}\Phi(\mathbf{x}_t)(\Phi(\mathbf{x}_t)^\mathsf{T}\hat{\Sigma}\Phi(\mathbf{x}_t))^{-1}\Phi(\mathbf{x}_t)^\mathsf{T}\hat{\Sigma}\right).$$

We can compute the approximated conditional mean and variance of the observation on $x \in \mathfrak{X}$ to be

$$\hat{\mu}_t(x) = \Phi(x)^\mathsf{T}\hat{\mathbf{u}}_t \quad \text{and} \quad \hat{k}_t(x) = \Phi(x)^\mathsf{T}\hat{\Sigma}_t\Phi(x).$$

Again, we prove a bound on the best-sample simple regret $r_T = \max_{x \in \mathfrak{X}} f(x) - \max_{t \in [T]} f(x_t)$. The evaluated inputs $\mathbf{x}_t = [x_\tau]_\tau^t$ are selected either by a special case of GP-UCB using the acquisition function

$$\alpha_{t-1}^{\text{GP-UCB}}(x) = \hat{\mu}_{t-1}(x) + \zeta_t\hat{k}_{t-1}(x)^{\frac{1}{2}}, \quad \text{with}$$

$$\zeta_t = \frac{\left(6(N - 3 + t + 2\sqrt{t\log\frac{6}{\delta}} + 2\log\frac{6}{\delta})/(\delta N(N - t - 1))\right)^{\frac{1}{2}} + (2\log(\frac{3}{\delta}))^{\frac{1}{2}}}{(1 - 2(\frac{1}{N-t}\log\frac{6}{\delta})^{\frac{1}{2}})^{\frac{1}{2}}}, \delta \in (0, 1),$$

or by a special case of PI using the acquisition function

$$\alpha_{t-1}^{\text{PI}}(x) = \frac{\hat{\mu}_{t-1}(x) - \hat{f}^*}{\hat{k}_{t-1}(x)^{\frac{1}{2}}}.$$

This special case of PI assumes additional information of the upper bound on function value $\hat{f}^* \geq \max_{x \in \mathfrak{X}} f(x)$.

For convenience of the notations, we define $\bar{\sigma}^2(x) = \sigma^2 \Phi(x)^{\mathrm{T}} (\Phi(\bar{\boldsymbol{x}}) \Phi(\bar{\boldsymbol{x}})^{\mathrm{T}})^{-1} \Phi(x)$.

Corollary 13 combines Lemma 2 and basic properties of the Wishart distribution [3].

**Corollary 13.** *Assume the matrix $\Phi(\bar{\boldsymbol{x}}) \in \mathbb{R}^{K \times M}$ has linearly independent rows. Then, $\hat{\boldsymbol{u}}$ and $\hat{\Sigma}$ are independent and*

$$\hat{\boldsymbol{u}} \sim \mathcal{N}\left(\boldsymbol{u}, \frac{1}{N}(\Sigma + \sigma^2(\Phi(\bar{\boldsymbol{x}})\Phi(\bar{\boldsymbol{x}})^T)^{-1})\right), \hat{\Sigma} \sim \mathcal{W}\left(\frac{1}{N-1}\left(\Sigma + \sigma^2(\Phi(\bar{\boldsymbol{x}})\Phi(\bar{\boldsymbol{x}})^T)^{-1}\right), N-1\right).$$

*For finite set of inputs $\boldsymbol{x} \subset \mathfrak{X}$, $\hat{\mu}(\boldsymbol{x})$ and $\hat{k}(\boldsymbol{x})$ are also independent; they satisfy*

$$\hat{\mu}(\boldsymbol{x}) \sim \mathcal{N}\left(\mu, \frac{1}{N}(k(\boldsymbol{x}) + \bar{\sigma}^2(\boldsymbol{x}))\right), \hat{k}(\boldsymbol{x}) \sim \mathcal{W}\left(\frac{1}{N-1}\left(k(\boldsymbol{x}) + \bar{\sigma}^2(\boldsymbol{x})\right), N-1\right).$$

The proofs of Lemma 3 and Theorem 4 in the paper directly follow Corollary 13 and proofs of Lemma 6, Theorem 11 in this appendix.

# D    Proofs for Section 4.3

We show that the simple regret with $\hat{x}_T^* = x_\tau, \tau = \arg\max_{t \in [T]} y_t$ is very close to the best-sample simple regret.

**Lemma 14.** *With probability at least $1 - \delta$, $R_T - r_T \leq 2(2 \log \frac{1}{\delta})^{\frac{1}{2}} \sigma$.*

*Proof.* Let $\tau' = \arg\max_{t \in [T]} f(x_t)$ and $\tau = \arg\max_{t \in [T]} y_t$. Note that $y_\tau \geq y_{\tau'}$. By Corollary 1, with probability at least $1 - \delta$, $f(x_\tau) + C\sigma \geq y_\tau \geq y_{\tau'} \geq f(x_{\tau'}) - C\sigma$, where $C = (2 \log \frac{1}{\delta})^{\frac{1}{2}}$. Hence $R_T - r_T = f(x_{\tau'}) - f(x_\tau) \leq 2C\sigma$.    □

# E    Experiments

For Plain and TLSM-BO with UCB in our experiments, we used the same $\zeta_t$ as PEM-BO.

In the following, we include extra experiments that we performed with PI acquisition function and matrix completion for the missing entry case in the discrete domains. The PI approach uses the maximum function value in the training dataset $\bar{D}_N$ as the target value. These results show that our approach is resilient to missing data. BO with the PI acquisition function performs similarly to UCB.