[Reviews · NeurIPS 2018]

Reviewer 1



The problem of hyperparameters tuning in GPs is indeed relevant. Unfortunately, I do not feel that the work is strong enough, in its current state, to provide clear enough results that other researchers could build upon. I am familiar with the analysis of GP-UCB and with Bayesian optimization. This paper addresses the problem of hyperparameters tuning in Bayesian optimization (e.g. with Gaussian processes). In practice, these hypers are often tuned by maximizing likelihood, at the expense of loosing theoretical guarantees. In this work, the authors propose to tune hypers using data previously gathered on similar (yet different, and independent) tasks and provide theoretical guarantees on the resulting estimators. They use these guarantees to provide examples of regret bounds of known acquisition functions using their techniques. Quality: The definition of simple regret (line 124) does not correspond to the definition from the literature (e.g. Bubeck & Cesa-Bianchi (2012): Regret analysis of stochastic and nonstochastic multi-armed bandit problems.). This quantity typically takes into account the error on the recommended arm (action, location to sample) after T rounds, while here it is taken w.r.t. the best sampled location. In this sense, the definition considered here feels weaker. How does this impact the theoretical results, i.e. Theorems 2 and 4? Questions regarding the case where X is compact: 1) It is assumed that \Phi are given. Does that mean that \Phi should be accessible/explicit? In this case, what would this mean for kernels that that have K = \infty? 2) An assumption is that the number of training points per previous dataset M \geq K. How does this impact on the choice of kernel given that it is possible that K = \infty (e.g. with the RBF kernel)? 3) It is also assumed that T \leq K. Again, how does this impact on the choice of kernel given that it is possible that K = 1 (e.g. linear kernel with X \subset \Real)? Lemmas 1 and 3 require that the size of the training dataset N \geq T+2. Similarly, Thm. 2 and 4 mention that \eta converge to a constant at rate 1/sqrt(N-T), implying the requirement that N > T. Why is it required that the size of the prior dataset is larger than the number of episodes? It is counterintuitive to me. Thm. 2 requires that N > M. I would appreciate some intuition on the reasons that motivates that prior dataset size is larger than the number of samples per sub-datasets. It is mentionned that the proposed approach is similar to the existing BOX algorithm, but that the proposed approach "directly improve on BOX by choosing the exploration parameters in GP-UCB more effectively" (lines 104-105). Where is this shown? Clarity: Though it is interesting in practice to consider missing observations, this is not the problem addressed by this work. Therefore, Sec. 4.1 could be alleviated from the related notation (everything revolving around \bar \delta_{ij}), as it is never used later in the paper. Elements are sometimes referred way earlier than their introduction, which makes the paper hard to follow. This is the case of "the problem shown in Fig. 1" (lines 47-48), which is only described later in Sec. 5. I guess that the approach referred to as "PEM-BO" in Sec. 5 is the proposed approach? It would be good to state this clearly. Overlall, the presentation of results is also hard to follow. For example: - Fig. 2a and 2c respectively show the learning curve for T = 10 and T = 5. What about Fig. 2b? - Legends in all subfigures of Fig. 2 should be increased for a better reading. - It is confusing to report experiments results in terms of rewards while theoretical results are provided in terms of regret. - There seem to be four approaches in Figs. 2d), 2e) and 2f), but only three approaches are introduced/dicsussed. Originality: The paper addresses a well-known and relevant problem, that is the tuning of Gaussian process hypers while preserving theoretical guarantees. The proposed approach is to tune the hypers using previously acquired observations of functions sampled from the same GP prior. It mentionned that the proposed approach is similar to BOX, but it is not clear how. The choice of literature to cite is sometimes odd. For example, citing the seminal UCB paper (Auer 2002) as an example of acquisition functions (lines 61-62). I would expect the followings: - PI: Kushner (1964) - EI: Mockus et al. (1978) - GP-UCB/CGP-UCB/KernelUCB: Srinivas et al. (2010), Krause & Ong (2011), Valko et al. (2013) Similarly, the references for GP-UCB (line 41) should be Srinivas at al. (2010) and Krause & Ong (2011). There may also be a connection with transfer learning that could be explored. Significance: This paper addresses an important problem, that is the need to tune GP hyperparameters while providing theoretical guarantees. Theorems 2 and 4 could be of possible interest outside this specific problem. These results are also used to update bounds of known acquisition functions (PI and GP-UCB) that would be using the proposed technique. However, considering the remaining questions regarding the unconventional definition of simple regret considered here, it is not clear how these new bounds would impact the literature. ====== Update after rebuttal: The only concern that I still have following the rebuttal is regarding the need to have N > T. It is strange to me and not very practical in real life. However, I understand that the theoretical analysis presented in this paper could contribute to the BO field. I updated my rating accordingly.

Reviewer 2



Detailed Comments: Summary of the main ideas: The authors propose a method based in a variant of Empirical Bayes for GP-based-BO to provide point estimates for the mean and kernel of the GPs. This method, which is essentially a sort of transfer learning method as the authors state, has the restriction that it needs a dataset of points to work. Authors provide a theoretical analysis of a regret bound by using their method and the GP-UCB and the PI acquisition functions. The paper is more formally written that other papers in the field which is a relief and it is, in my humble opinion, very necessary. It also provides a theoretical analysis, which is also rare in the field, and also necessary. Related to: Kim, Beomjoon, Leslie Pack Kaelbling, and Tomás Lozano-Pérez. "Learning to guide task and motion planning using score-space representation." Robotics and Automation (ICRA), 2017 IEEE International Conference on. IEEE, 2017. BOX algorithm that is the base of the methods that this paper suggest to provide estimates of mean and covariance matrices over a discrete domain. (This paper generalizes this for a continuous domain). Robbins, Herbert. "An empirical Bayes approach to statistics." Herbert Robbins Selected Papers. Springer, New York, NY, 1985. 41-47. The empirical Bayes methodology that suggest fixed values instead of a distribution to sample hyperparameters. They adopt a variant of empirical Bayes in this paper. GP-UCB and PI acquisition functions, along with an easy to read tutorial of BO, that this paper analyze are described in Brochu, Eric, Vlad M. Cora, and Nando De Freitas. "A tutorial on Bayesian optimization of expensive cost functions, with application to active user modeling and hierarchical reinforcement learning." arXiv preprint arXiv:1012.2599 (2010). Strengths: Theoretical and empirical analysis of the method. Bayesian statistics applied in a way that it has not been applied before for BO priors. Very formally written. Very well organized. Weaknesses: Difficult for non BO experts. (This weakness can be overcome by the suggestions that I provide in this review). Does this submission add value to the NIPS community? : Yes it does. I think that I have only read one or two papers in the BO field that provide such an excellent theoretical and empirical analysis. Moreover, theoretical results are critical in BO, and it does not only study the GP-UCB AF, which has been theoretically analyzed before, but also PI. The method is sound and based in solid bayesian statistics. It frees the BO user for having to select the kernel, which is a heavy burden sometimes and something ironical in a sense. If BO is used for AutoML it is ironical that it has to be auto-tuned as well. Personally, I think that these applications of bayesian statistics and theoretical analysis of BO should be more studied. But the key fact of the paper is that it is not only theoretical but it also provide a good empirical analysis. For all these reasons, I strongly recommend this paper and state that it does add value to the NIPS community. Quality: Is this submission technically sound?: Yes, this way of estimating the mean and kernel has not been done before. Are claims well supported by theoretical analysis or experimental results?: Yes, it is the main strength of the paper. Is this a complete piece of work or work in progress?: Complete piece of work. Are the authors careful and honest about evaluating both the strengths and weaknesses of their work?: Clarity: Is the submission clearly written?: It has some issues for non experts. The paper assumes strong knowledge of BO and bayesian statistics. This may be a burder for non experts. I suggest adding some images, for example. An image of an estimation of the mean in 1D given some data, or a flow diagram with the offline and online phases. This would add some friendly style that lots of readers of this conference would appreciate. Also, all the variables must be explained, for example, in line 120, the different indexes i and j are not explained, this may be confusing for some readers. Special cases of the PI and GP-UCB are not described and their equations are not given numbers. These expressions are so easy to explain and so necessary for not experts that if they are not explained they may abandon to read the paper further. It would be a great pity. On the other hand, proofs of the papers add clarity to some settings as the hyperparameter setting of the GP-UCB that are great and are not also common in BO papers. Algorithm for Meta-BO is very clear and helpful. Is it well organized?: Yes it is. It is very well organized. All sections introduce the content of posterior ones. Algorithms are brillianty referenced. One thing that needs to be added is a conclusions section. Does it adequately inform the reader?: If clarity issues that I have stated in Question 1 of Clarity are solved. Then, it does adequately inform the reader. Originality: Are the tasks or methods new?: There are extensions of well-known techniques (BOX and Empirical Bayes). Is the work a novel combination of well-known techniques?: Yes it also is. Because it combines the extensions. Is it clear how this work differs from previous contributions?: Yes it is. It is different to transfer learning BO techniques and usual BO initialization of the mean and variance. Is related work adequately cited?: Yes it is. Lots of references are provided. Significance: Are the results important?: Yes they are. This methodology is different from previous ones and seeing theoretical and empirical results of it is a guarantee of the success. Are others likely to use the ideas or build on them?: The paper is a bit hard to read, if the style is frendlier, I am sure that it is going to be sused. Does the submission address a difficult task in a better way than previous work?: Yes, in a sense. Using empirical Bayes and leaving free the kernel is a good idea. Does it advance the state of the art in a demonstrable way?: Yes it does, it demonstrates from an empirical and theoretical way. Does it provide unique data, unique conclusions about existing data, or a unique theoretical or experimental approach?: Yes, proofs in the supplementary material complement the exposition of the manuscript and provide an unique theoretical approach. Arguments for acceptance: The theoretical and empirical analysis for a BO method that this paper performs is really unusual in BO. Moreover, the paper has coherence, the methodology is sound and solid and it advances the state of the art. I strongly suggest this paper for acceptance. This paper is also written with an excellent style, rarely found in computer science papers. I had a hard time to find typos, everything is written very carefully. Authors have a really strong knowledge of the field. Arguments against acceptance: If clarity issues are not solved, the paper is difficult to read. The style is great but, again, the paper is difficult to read and understand. Variables and equations are not always explained. I suggest to provide more intuitive explanations. Typos: Line 17, provide here the Rasmussen reference. [37]. Line 82, instead of writing k, write $k$. More detailed comments and suggestions: Congratulations to the authors, please make it easier to read for newcomers and this paper will be a success.

Reviewer 3



This paper proposes to estimate the Gaussian process (GP) mean and covariance function using previous observations generated by functions sampled from the GP, and then use this estimated GP prior to conduct Bayesian optimization on a new test function, which is also assumed to be a sample path from the same GP. The main contribution of this paper is the derivation of the (simple) regret bounds of such a procedure using a special setting of the Upper Confidence Bound (UCB) or Probability of improvement (PI) policies, on both finite and compact domains. The idea of such transfer learning seems to be a formalization of the recently proposed BOX algorithm [24] and that of Google vizier [16] (except here is not sequential), and is generalized to continuous domains. The generalization is simply degenerating the GP to parametric Bayesian linear regression in a nonlinear space transformed using given basis functions (in experiments, the basis functions are learned using a single-layer neural network by minimizing the average loss over all training sets). The regret bounds seem to be novel. I would like to know how this theoretical analysis is related to, e.g., that of Srinivas et al. (2010) [42]. Though one being simple regret and the other cumulative regret, you both analyzed the UCB policy and the notion of information gain is involved in both. The paper is well-written and easy to follow. I'm a little concerned about the experiments. In the first simulated experiments, where the true GP is known, the PlainUCB using the true GP appears to be significantly worse than your algorithm, which uses the estimated GP. I would expect the two lines in Figure 2(d) to be very close if there is enough training data to estimate the GP (or at least PlainUCB shouldn't be worse). Perhaps this is only due to lack of repetitions. Could you conduct more trials to reduce the variance and see if you can verify these? In the second and third experiments, the comparison seems unfair for PlainUCB since an inappropriate (as also noted by the authors) square exponential kernel was chosen. In practice it's true that we may not know what kernel is appropriate, but at least we could avoid using a kernel known to be inappropriate, unless a better one is not known. Another comparison that I think would also be interesting is what if you also tune the hyperparameters of the PlainUCB kernel using the training data by empirical Bayes? In Figure 2(f), how come the line of PemboUCB started off much higher than others? Shouldn't they all be initialized using the same set of seeding points? And "rand" (I suppose it means random?) is better than the other two baselines?